

# Impacts of science on society and policy in global river basins

Shuanglei Wu[1], Yongping Wei[1]

[1] School of the Environment, the University of Queensland, Brisbane, 4072, Australia.

*Correspondence to*: Yongping Wei (yongping.wei@uq.edu.au)

**Abstract.** Radical transformations of knowledge development are required to the sustainability issues in the Anthropocene. This study developed a framework to understand the internal structures of knowledge development with two dimensions: Degree of Multidisciplinarity and Degree of Issue-connectivity. Examining the knowledge development in 72 river basins globally from 1962 to 2017, it was found that the knowledge systems were characterized by intensified issues studied and low and ungrown disciplinary engagement. Evaluating these structural characteristics against 6 impact indicators on society and

policy, over 90% of rivers were found to had knowledge structures that strongly linked to societal impacts whereas only 57% were to the policy. Analysis were further conducted to find that about 35% of rivers mostly in Asia, Africa, and South America were prone to knowledge structures that had limited capacities to effectively address negative environmental impacts and resource depletions issues. Improving multidisciplinary research is the key to transforming the current knowledge structure to support more sustainable river basin development.

## 1 Introduction

Science is often called upon to provide solutions to societal problems and is also a common ingredient of policy making. However, the exponential development of science and technology with its irreversible environmental and social side effects is pushing the Earth's safe operating space close to its planetary boundaries (Steffen et al., 2015; Brey, 2018). Therefore, radical transformations of knowledge (science and technology) development are required to meet the rapidly changing societal needs

in the Anthropocene (Norström et al., 2020; Hakkarainen et al., 2022).

Advancing knowledge management and assessment is a key to radical transformations of knowledge development. Current studies on knowledge management and assessment mainly adopt a range of input-output indicators (e.g., R&D inputs, number of scientific papers and citations) and several evaluation tools (e.g. bibliometrics, case studies, and benchmarking) (Penfield et al., 2013). They tend to focus on the quality of scientific output, i.e., the "credible, legitimate, and relevant" criteria of

"good science" (Cash et al., 2003; Posner and Cvitanovic, 2019). While these studies have provided fruitful insights on the ways in which science has produced impacts, they have limited applicability beyond the case study areas, and largely overlook the internal structural dynamics of the knowledge system where different disciplinary knowledge interact to address increasingly complex issues that may significantly impact the society and policy-making (Weichselgartner and Kasperson, 2010; Hakkarainen et al., 2020). Without understanding and addressing the possible structural failure of knowledge



development, we would not be in a position to direct knowledge transformations (Wu et al., 2021; Wei et al., 2022; Newig
and Rose, 2020).

This study developed a framework to understand the internal structures of knowledge development and evaluate the impacts
of these structural dynamics on society and policy to structurally reconfigure the knowledge systems for addressing complex
sustainability issues. The framework was empirically applied in the knowledge development of 72 river basins across the world

from 1962 to 2017. The river basins were chosen as an example as they are logical spatial units to understand the water cycle
and its interactions with the integrated Earth System (Warner et al., 2008). Further, water is a key input for almost all economic
activities with broad impacts on both society and policy, and better management of water to achieve sustainable development
goals has become a global concern (Rodríguez et al., 2021).

## 2 A network-based framework to measure the structure of knowledge systems

A knowledge system can be understood as a dynamic network with different disciplines "*knitting, weaving and knotting
together into an overarching scientific fabric*" (Latour, 1987; Shi et al., 2015). Built on our previous framework (Wei and Wu,
2022; Wu et al., 2021), we define knowledge as a network system, within which disciplinary knowledge and issues studied
co-evolve and feedback to each other. The interactions between them form the structure of a knowledge system (Shi et al.,
2015; Coccia, 2020; Newman, 2003), determining the knowledge system's functionality (i.e. impact) (Von Bertalanffy, 1968;

Huttenhower et al., 2012; Sayles and Baggio, 2017).

We use two dimensions to capture the topological structure of a knowledge system (Wasserman and Faust, 1994; Borgatti,
2005; Zeng et al., 2017). First is the Degree of Multidisciplinarity (DM) that indicates the proportions of disciplines engaged
in different issues and is measured as the density of the discipline-issue network. This dimension recognizes the importance
of disciplinary diversity in sustainability issues (Norström et al., 2020; Cockburn, 2022; Stirling, 2007). Second is the Degree

of Issue-connectivity (DI). It indicates how different issues are studied in an interactive manner and is measured as the degree
centrality of the issue network. This dimension recognizes the increasing complexity in sustainability issues and the importance
of understanding these issues in an interactive manner (Burmaoglu et al., 2019; Okamura and Nishijo, 2020).

To compare the relative differences of DM and DI among rivers, the z-scores for DM and DI ($x_{z,k}$) in any river k are calculated
by subtracting the means ($\overline{x_k}$) then divided by the standard deviations ($s.d.x_k$) of all rivers. Four types of knowledge structures

are defined (Fig. 1): A) Integrated knowledge structures ($DM_{z,k} > 0$, $DI_{z,k} > 0$) with diverse disciplines engaged in
interconnected issues; B) Issue-driven knowledge structures ($DM_{z,k} < 0$, $DI_{z,k} > 0$) with limited disciplines engaged in
interconnected issues; C) Fragmented knowledge structures ($DM_{z,k} < 0$, $DI_{z,k} < 0$) with limited disciplines engaged in
isolated issues; and D) Discipline-driven knowledge structures ($DM_{z,k} > 0$, $DI_{z,k} < 0$) with diverse disciplines engaged in
isolated issues. An integrated knowledge structure is considered to be ideal in studying highly interacted issues with diverse

disciplines; while a fragmented structure is at the other end of the spectrum that both issues and disciplines are in silos. An
issue-driven knowledge structure tends to provide disciplinary-specific solutions for interactive issues, which are often cost-





effective in the short term but may lead to unintended or unexpected outcomes in the long term due to the narrow perspective of disciplines. A discipline-driven knowledge structure tends to provide trans-disciplinary solutions for key issues of focus, which are often not cost-effective in the short term as it often takes a long time and requires large investments to find a solution,

but more sustainable in the long term. In time, knowledge development may demonstrate different structural pathways, for example moving from the under-developed fragmented structure to a discipline-driven structure, and/or from an issue-driven structure towards an integrated one.

We further expand our framework to evaluate the impacts of knowledge development. The commonly recognised triple-bottom-lines framework is adopted to define the impact of the knowledge system on society (Reyers and Selig, 2020), which

includes the social (SO), economic (EC) and environmental (EN) dimensions. We then uniquely define the impacts of the knowledge system on policy according to the whole-of-system characteristics in natural resources management. It includes resource availability (RA), resource utilization (RU), and governance capacity (GC) (Wei et al., 2018; Ostrom, 2009). Resource availability refers to the supply capacity of natural resources, resource utilization reflects the extent to which a resource is used, and governance capacity indicates the government's regulation on the supply and demand of a resource.

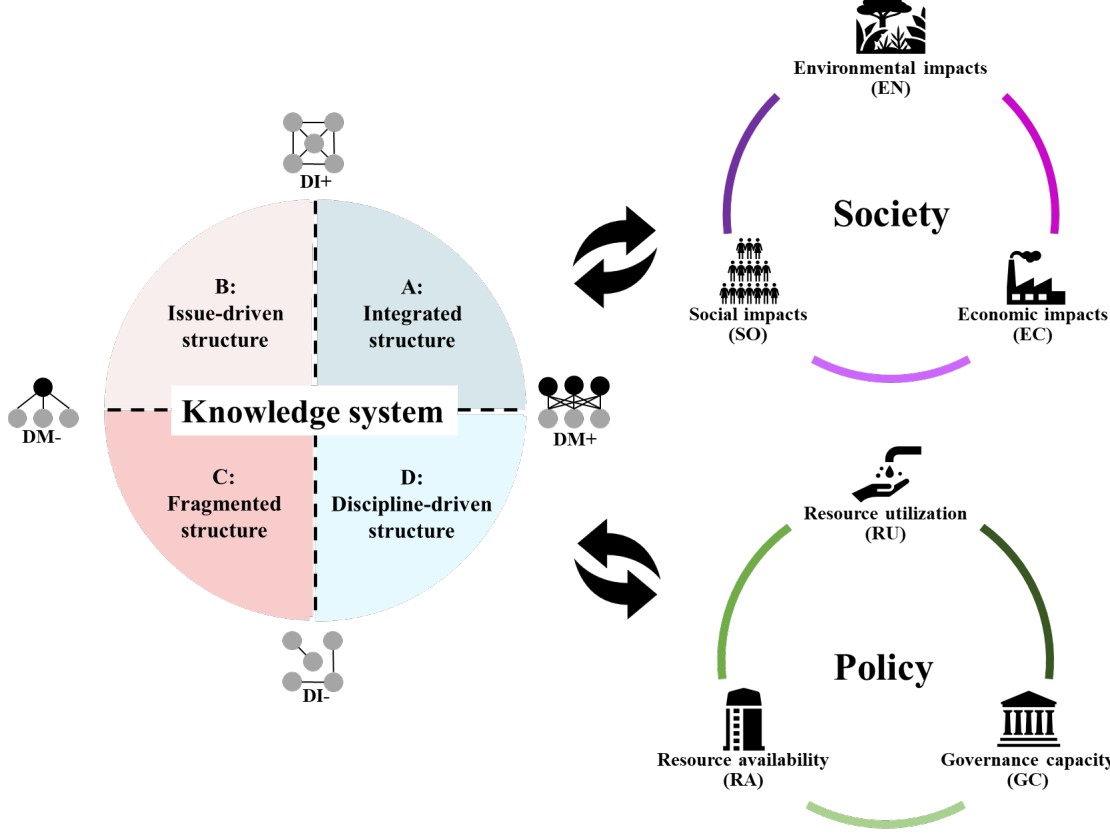


**Figure 1: A framework to understand the knowledge system and its impacts with society and policy in natural resources management.**





# 3 Methods

## 3.1 Data collection and processing

*The river basin knowledge system*

The river basin knowledge system was represented by peer-reviewed articles indexed in the Web of Science (WoS). Archiving over 21,000 high quality scholarly journals, the WoS is one of the largest databases that document knowledge development since 1900. It provides up-to-date, consistent classifications of knowledge under the Master Journal List (https://mjl.clarivate.com/home), which classifies articles according to their source journals into 254 disciplines under five

research areas: Arts & Humanities, Life Sciences & Biomedicine, Physical Sciences, Social Sciences, and Technology (Clarivateanalytics, 2018). Articles with "drainage basin" OR "river basin" OR "valley" OR "hydrographic basin" OR "watershed" OR "catchment" OR "wetland" in their Titles, Abstracts and Keywords sections were collected.

Four types of information were extracted from each article: disciplines, year of publication, keywords, and river basin studied. The discipline and year of publication for each article were automatically assigned based on their source journals. For journals

with multiple disciplines, only the first, most dominant discipline was assigned. A total of 215 disciplines were identified (see Table A1 for a full list). The keywords were extracted, filtered, and tokenized from the Titles, Abstracts and Keywords sections of the articles using the Natural Language Processing (NLP) module in the Derwent Data Analyzer (https://clarivate.com/derwent/zh-hans/solutions/derwent-data-analyzer-automated-ip-intelligence/). Those keywords related to the methodologies of the articles were removed and the remaining were regrouped manually into the 94 issues that broadly

represent major topics of river basins research and management (e.g., agriculture, pollution, climate change, see Table A2 for a full list and also Wei and Wu (2022) for more details on grouping of the keywords).

Each article was also assigned a river basin to which it was used as a case study. All articles without a clear indication of case river basins and duplicated articles were removed. Initially, the top 100 most published river basins were selected. Removing those with ambiguous river basin names and those river basins with unenclosed coastal shorelines that lack country-level data,

a total of 72 river basins were finally selected. They cover major river basins in the world.

Discipline-issue networks were then constructed based on the co-occurrence principle (Noyons, 2001; Callon et al., 1983). For each article, keywords were connected to the corresponding disciplines assigned, and the number of articles was counted as the weights of connections. Totally, 165,044 discipline-issue connections were identified.

The discipline-issue network and the issue network were constructed for each of the 72 river basins for analysis.

To quantify the structure of the knowledge system, the DM measure was calculated by the density of the discipline-issue network, which was the ratio between the actual number of connections between different keywords and disciplines to the total possible connections in the network (Eq.1). The higher the DM, the more disciplines were involved and the more multidisciplinary the knowledge system was. For any node d (a specific keyword or discipline) in the network:

$$DM = \frac{\sum_i \text{no. of existing connections between } d_i}{\text{Maximum potential connections}} \qquad \text{(Eq.1)}$$





Then, the discipline-issue network was projected into an issue network so that only connections between issues were analyzed. This was achieved by following the co-occurrence principle again. The DI measure was calculated by the degree centrality in the issue networks (Wasserman and Faust, 1994; Borgatti, 2005), which was the average value of the connections for each keyword k in the issue network (Eq.2). The greater the DI, the more interacted the keywords are and the more centralised the knowledge system is.

$$DI = \frac{\sum_j \text{adjacent edges connected to } k_j}{\text{total number of } k} \qquad (Eq.2)$$

*Indicators to represent society and policy*

We chose the indicators for society and policy based on the following principles: 1) expressed quantitatively; 2) reflecting system processes rather than end-states; 3) data availability; and 4) specific focus on impacts related to water resources. For the society, the economic impact was defined by water productivity, which was the economic values generated by water resources use. The societal impact was represented by populations to show the total size of human demand for water resources, and the environmental impact was an negative indicator of water stress. Greater water stress indicated greater negative impacts on the environment. For the policy, resource availability was represented by the percentage of cultivated land. While precipitation or runoff is commonly recognized as a key indicator for water resource availability, we selected cultivated land as its change is more influential on water resources management. It was an negative indicator, which means that increasing cultivated land increases water resources use, thus reducing the availability of water resources. Resource utilization was represented by total freshwater withdrawals to indicate the size of water uses, and governance capacity was represented by a normalised Government Effectiveness Index that indicated the regulation effect of policy.

Data on the indicators for both the society and policy were collected from the AQUASTAT database by the Food and Agriculture Organization (FAO), the World Bank, the Socioeconomic Data and Applications Centre (SEDAC) by NASA. In particular, population and water withdrawal data have been improved by Yan et al. (2022) combining FAO, SEDAC databases and local government archives with extended temporal and spatial scales, which was adopted in this study. The chosen indicators with brief descriptions and corresponding temporal and spatial scales as summarised in Table 1:

**Table 1: Summary of indicators on society and policy**

| Indicator | Description | Data source | Spatial resolution | Temporal resolution |
|---|---|---|---|---|
| **Society** | | | | |
| Social impact | Total population (1000 people): A measure of the size of society that defines the human demands of water resources. Higher value indicates greater human water demand. | SEDAC, expanded and adjusted based on Yan et al. (2022) | Gridded data at 1 km. | Yearly from 1962 to 2017. |





| Economic impact | Water Productivity (constant 2015 US$ GDP per cubic meter of total freshwater withdrawal):<br>A monetary measure of the efficiency of water resources use.<br>Higher value indicates greater economic efficiency. | AQUASTAT, The World Bank | Country level. | Every five years from 1962 to 2017. |
|---|---|---|---|---|
| Environmental impact (negative indicator) | Water Stress (% of freshwater withdrawal to available freshwater resources):<br>A percentage measure taking into consideration of the environmental impacts of water use, also an indicator of the Sustainability Development Goal (SDG) 6.4.2.<br>Higher value indicates greater stress and worse environmental condition. | AQUASTAT | Country level. | Every five years from 1962 to 2017. |
| **Policy** | | | | |
| Resource availability (negative indicator) | Percent of total country area cultivated (% of cultivated area to country area):<br>A percentage measure of the land use that defines the biophysical demand of water use.<br>Higher value indicates lower availability. | AQUASTAT | Country level. | Every five years from 1962 to 2017. |
| Resource utilization | Total freshwater withdrawal ($10^9$ m$^3$/yr):<br>A measure of water use.<br>Higher value indicates greater use. | AQUASTAT, expanded and adjusted based on Yan et al. (2022) | Gridded data at 1 km. | Yearly from 1962 to 2017. |
| Resource governance | Government Effectiveness Index (normalised percentile index between 0 and 100):<br>A composite index measuring the quality of policy formulation and implementation | The World Bank | Country level. | Yearly from 1996 to 2017. |



| | | | |
|---|---|---|---|
| based on survey data from households, business firms, public organizations and NGOs. Higher value indicates better governance. | | | |

To aggregate the different spatial scales of data to a river basin scale, the boundaries of the 72 river basins were determined.
26 river basins boundaries were identified as transboundary and collected from the Transboundary Waters Assessment Programme (TWAP). The basin boundaries of the remaining 46 river basins located entirely within single countries were collected from corresponding national records (e.g. the U.S. Geological Survey, the Murray-Darling Basin Authority). For each transboundary river basin, a basin area ratio was calculated as the weighted proportion of river basin area and populations for each country within the boundary of the river basin. The country-level indicators were then multiplied by the basin area
ratio, and then aggregated by the average values for all spanning countries in the basins. For river basins located entirely within single countries, the country-level indicators were assumed to be the same within the basin boundaries. All gridded level indicators were clipped based on the basin boundaries and averaged across the basin area using ArcGIS Pro 3.0.

Finally, missing values at country levels in time were imputed by linearly interpolated the missing values based on the regression relationship between the existing values in the time series. For the Government Effectiveness Index which was not
available before 1996, values were assumed to the same as the first available year.

A study period from 1962 to 2017 at five-yearly intervals was used. This study period was chosen to reflect the history of water resources development closely tied with rapid socio-economic development, environmental deterioration, and a governance system transitioning from technocratic, top-down control to collaborative, integrated management (Molle, 2009). Also, there was limited data availability on society and policy at a global scale before 1962 and after 2017.

**3.2 Analysis approaches**

*Time trend analysis*

The Mann-Kendall test was used to test if there exist statistically significant, monotonic increasing/decreasing trends in the time series for the knowledge system and its impacts (Mann, 1945; Kendall, 1975). Significant trends were identified with two-sided t-test with p value < 0.05.
Theil-Sen's slopes (Sen, 1968) were then used to calculate the magnitude of the trends as Eq.3:

$$d_{Sen} = \text{median} \left( \frac{x_j - x_i}{j - i} \right) \text{ for } 1 \leq i < j \leq n \qquad \text{(Eq.3)}$$

where x is the indicator value in time series, $i$ and $j$ are time points and n is the total number of data points. As non-parametric measures, these analyses are insensitive to outliers and autocorrelations in the time series, and do not require data that satisfy the normality assumption, thus providing robust measures of the time trends for indicators with varying scales and limited data
amounts (Wang et al., 2020; Fernandes and G. Leblanc, 2005)

*Measuring the knowledge system impacts*



To compare the impacts on society and policy with different scales, normalisation was conducted over the time series of the knowledge indicators (i.e., DM and DI), and the society and policy indicators within the range of 0 and 1 (Eq.4).

For each year n and river basin k, and any knowledge, societal, or policy indicator x:


$$\text{Normalised indicator } x'_{n,k} = \frac{x_{n,k} - x_{k,min}}{x_{k,max} - x_{k,min}} \qquad \text{(Eq.4)}$$

Generalized linear regression models were then used to quantify the relationships between the normalised society and policy indicators as dependent variables and the knowledge system indicators as independent variables using Eq.5-6:

$$\text{Society IND}'_{i,k} = \alpha_{i,k} \times DM'_{i,k} + \beta_{i,k} \times DI'_{i,k} + \varepsilon_{i,k} \qquad \text{(Eq.5)}$$

$$\text{Policy IND}'_{j,k} = \alpha_{j,k} \times DM'_{j,k} + \beta_{j,k} \times DI'_{j,k} + \varepsilon_{j,k} \qquad \text{(Eq.6)}$$

where $\alpha$ and $\beta$ are the normalised coefficients representing the partial influences to which DM and DI have for river basin $k$ relate to a particular society indicator $i$ or a particular policy indicator $j$, and $\varepsilon$ is the random error terms capturing the biased values.

Models that failed to pass the two-sided t test with p value > 0.05 and/or with adjusted $R^2$ < 0.3 were rejected (Royston, 2007; Ratner, 2009). We recognized that the society and policy indicators can be influenced by a wide range of factors. Therefore,

these statistical models were not developed for causal inferences. Rather, we focused on the comparative impacts of the knowledge system to identify better knowledge system for different river basin biophysical and socio-political contexts.

***Determining the patterns of knowledge impact***

To identify the different interacting patterns between knowledge and society, and between knowledge and policy, the river basins were grouped based on their regression coefficients ($\alpha$ and $\beta$) for the society and policy indicators respectively. Firstly,

river basins with more than two statistically non-significant linear models regarding the three society indicators, and those regarding the three policy indicators were grouped separately. These river basins were identified to have knowledge systems with unclear impact patterns. Secondly, the remaining river basins were grouped using agglomerative hierarchical clustering (AHC) based on the Euclidean distances and Ward's agglomerative criterion, which was chosen as it was less prone to the randomness of clustering initiation and provided stable groupings of rivers (Murtagh and Legendre, 2014). Rivers were

clustered based on the six coefficients in the linear models with the society indicators (i.e., $\alpha$ and $\beta$ for social, economic, and environmental impacts), then grouped separately based on the six coefficients in models with the policy indicators (i.e., $\alpha$ and $\beta$ for resource availability, utilization, and governance). The number of clusters was chosen as 2 for the society and policy clustering respectively, which was determined by maximizing the sum of square errors between different groups and minimizing the errors within groups.

***Optimizing the knowledge system for its impacts***

We represented the four types of knowledge-impact relationships as the average coefficients ($\alpha_{avg}$, $\beta_{avg}$, $\varepsilon_{avg}$) for the corresponding linear models of the rivers in each knowledge-impact pattern group. These relationships were then used as the objective functions for multi-objective optimizations using a NSGA-II genetic algorithm (Deb et al., 2002; Coello coello et





al., 2020) to identify the optimum DM and DI values ($DM'_{opt}$, $DI'_{opt}$) that simultaneously achieve the objectives specified in Table 2.

The NSGA-II algorithm was selected because it searched for the global Pareto optimality for the multiple counteractive objectives in this study (Edgeworth, 1881; Deb and Gupta, 2005). It provided a set of effective solutions that were at least as good as other possible solutions for each objective and strictly better for at least one objective (Halffmann et al., 2022). Combining random numbers and information from previous search interactions over the whole of potential solution points, this algorithm has been effectively used to solve multi-objective problems, particularly in engineering and decision-making optimization (Marler and Arora, 2004). 100 pairs of potential DM and DI values were randomly generated initially and modelled over 1000 iterations to search for the optimum values. Finally, we evaluated the trade-offs and synergies of different objectives achieved by different optimized DM and DI values to recommend tailored management strategies for future knowledge system development.

**Table 2: Optimization objectives for knowledge-impact relationships**

| Knowledge-impact relationships | Optimization objectives |
|---|---|
| Society impacts (each of EC, SO, EN) = $\alpha_{avg} \times DM'_{opt} + \beta_{avg} \times DI'_{opt} + \varepsilon_{avg}$ | Maximize Economic impacts (EC); Maximise Societal impacts (SO); and Minimise Environmental (EN) impacts. |
| Policy impacts (each of RU, GC, RA) = $\alpha_{avg} \times DM'_{opt} + \beta_{avg} \times DI'_{opt} + \varepsilon_{avg}$ | Maximize Resource Utilization (RU); Maximise Governance Capacity (GC); and Minimise Resource Availability (RA). |
| **Subject to the following boundary conditions** | |
| $0 \le DM'_{i,k}$, $DI'_{i,k} \le 1$ | |

The above analysis were conducted using R version 4.2.3 with the following packages: "igraph" (https://igraph.org/r/), "imputeTS" (https://cran.r-project.org/web/packages/imputeTS/index.html), "Stats" (https://www.rdocumentation.org/packages/robustbase/versions/0.95-0), "factoextra" (https://cran.r-project.org/web/packages/factoextra/index.html), and "nsga2R" (https://cran.r-project.org/web/packages/nsga2R/index.html).

## 4 Results

### 4.1 The knowledge systems with intensified issues studied and low and ungrown discipline engagement

The knowledge systems of the 72 river basins were characterized by a limited increase of engagement of scientific disciplines, but intensifying interactions among issues studied. 47% of the river basins had positive temporal trends for DM but only 8 were statistically significant ($p < 0.05$), most of which were located in Asia (e.g., the Nakdong River, and the Yangtze River). About 40% had negative Sen's slopes, of which only 9 were statistically significant, spreading across North America, Europe,



and Oceania. Moreover, both the average significant positive and negative Sen's slopes only varied between 0.02% and 0.05% per 5 years, with clear stabilization of the absolute DM values between 0 and 0.25 for all river basins in 2017. Multidisciplinary research for global river basin studies were highly constrained within the biophysical disciplines, with over 70% of interactions identified among the Environmental Sciences, Water Resources, Ecology, Multidisciplinary Geosciences, and Marine & Freshwater Biology. Only about 10% of interactions were contributed by social sciences such as Human Geography, Economics, and Management (Fig. 2a, Table A1).


On the other hand, all river basins demonstrated statistically significant increasing trends for DI ($p < 0.05$). The top 5 river basins with the greatest positive trends were the Great Lakes, the Mississippi River, the Yangtze River, the Nile River, and the Chesapeake Bay, with an average Sen's slope of 6% increase per 5 years; which was about 12 times greater than the bottom 5 river basins. The Murray-Darling River had an increasing trend of 3.8% per 5 years as the only river studied in Oceania, followed by the European river basins with an average increasing trend of 2.7%. 50% of the river basins had absolute DI values between 20 and 40 (i.e. the average number of issue interactions in the knowledge system) and the highest DI value reaching nearly 80 (i.e., the Great Lakes) in 2017. While issues on ecological degradation and restoration, and pollution and treatments comprised about 40% of total connections, research issues were more diversely distributed among management and control, agriculture and irrigation, flood and drought management, climate change and population (Fig. 2b, Table A2).


Classifying the knowledge structures of river basins based on their average DM and DI values indicates that 35% of the river basins had fragmented knowledge structures with low DM and low DI, mostly in Asia. 25% river basins had integrated knowledge systems with relatively high DM and DI values, including the Murray-Darling River, the Colorado River, the Amazon River, the Nile River, and most of the European rivers. Most of the North American rivers were discipline-driven, whereas the European rivers tended to have integrated knowledge systems (Fig. 2c).







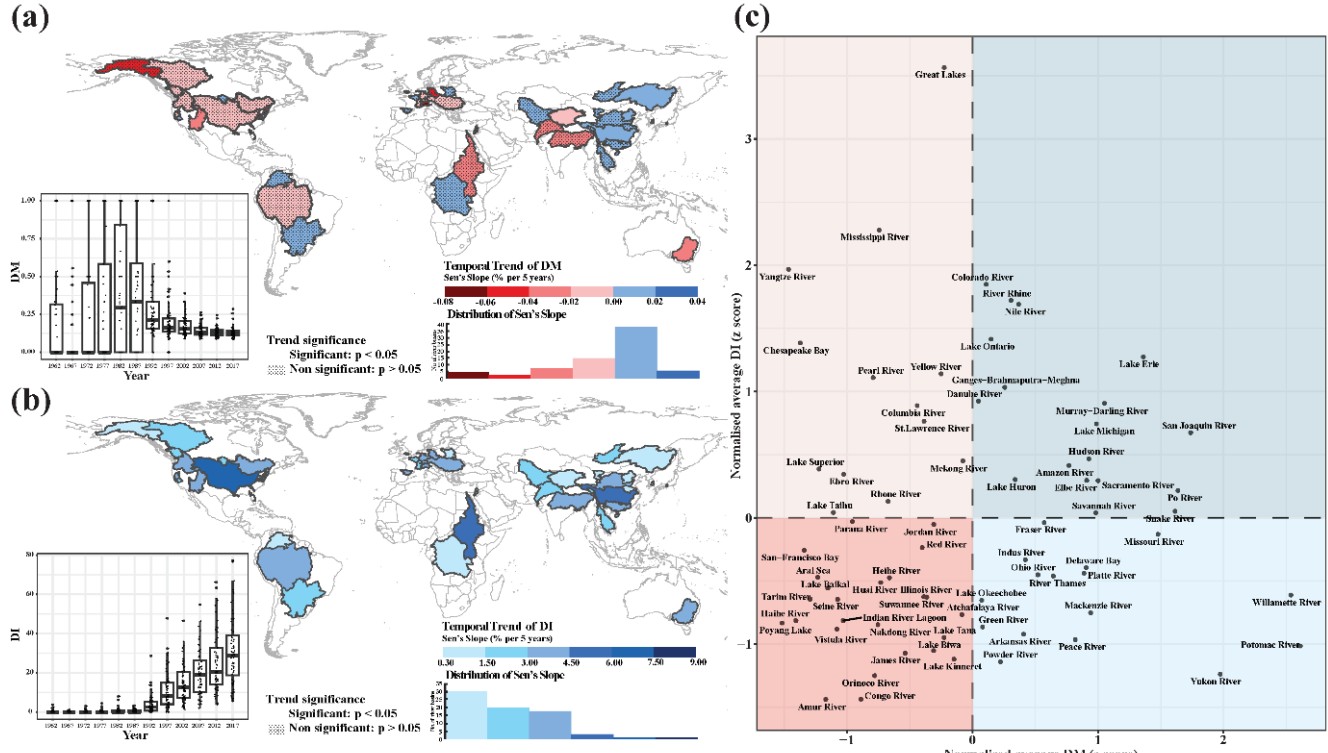

**Figure 2: (a)** The temporal trends (Sen's slope) and the absolute values (in inset) of the Degree of Multidisciplinary (DM) for the 72 river basins; **(b)** the temporal trends (Sen's slope) and the absolute values (in inset) of the Degree of Issue-connectivity (DI) for the 72 river basins; and **(c)** the knowledge system classification for the 72 river basins by their normalized average DM and DI. Dots in the boxplots indicate individual DM and DI values, the box boundaries indicate the 25th and 75th percentiles, the centre line indicates median values, and the whiskers indicate the 1.5 times of the interquartile range.

### 4.2 Unequal development of society and policy in global river basins

We then examined the development of the society and policy indicators in the 72 river basins, comparing the relative proportions among the society indicators and among the policy indicators, respectively. African and South American rivers had the greatest absolute SO increases of 25% per 5 years on average, which comprised over 60% of their relative proportions (except Lake Tana) in all society indicators. In additions, the EN increased the most for the South American (13.7%). Asian rivers had over 50% of their relative proportions contributed by SO and less than 20% by EC. Yet they also had the greatest absolute EC increases at over 60% on average, mostly by the Yangtze River, the Peral River, and the Yellow River (100%) whereas other rivers like Ganges-Brahmaputra-Meghna Basin, the Mekong River, and the Jordan River only increased by 10%. Most of the European and North American rivers had relatively stable proportions with low SO (0-40%), balanced EC (30-60%) and EN (50-70%), with European rivers having the least absolute increase in SO at less than 3% and all European and North American rivers had decreasing trends (-2.4% and -1.2% on average) in EN  (Fig. 3a, Fig. B1).

Lake Tana in Africa demonstrated the greatest relative proportions in RA, whereas the greatest absolute increases were observed for the South American (9.1%) rivers. The African, European, North American, South American rivers, and the





Murray-Darling Basin in Oceania had similarly lower relative proportions of RU (0-40%) and higher GC (50-100%). The African rivers had the greatest absolute RU increase at 43%, whereas only 10 rivers showed significant decreases at an average rate of -4.7%, all of which in North America or Europe. Furthermore, non-significant trends in GC were observed for over 60% of the rivers. Although the Asian rivers had only 20-50% relative proportions in GC, a significant increase of 2.4% per 5 years on average was identified (Fig. 3b, Fig. B2).

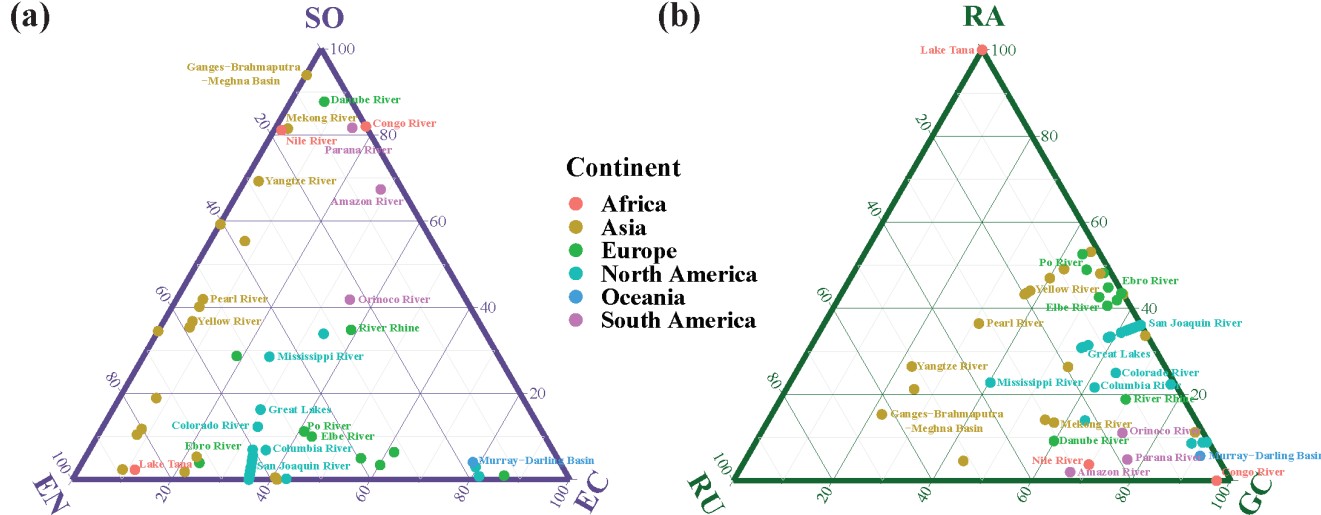


**Figure 3: (a) The relative average values of the social (SO), economic (EC), and environmental (EN) indicators; and (b) the relative average values of the resource availability (RA), utilization (RU), and governance (GC) indicators for the 72 river basins. Only the top 5 most published river basins in Asia, Europe, and North America, and all rivers in Africa, South America, and Oceania were labelled.**

**4.3 The knowledge structures linked to society more strongly than to policy**

The structural characteristics of the knowledge systems had been strongly linked to the society indicators with over 90% river basins had acceptable regression model fits, but much weaker with the policy indicators as only 41 river basins had two or more linear models that validated the relationships between their knowledge systems and the policy (adjusted $R^2 > 0.3$, statistical significance $p < 0.05$).

69% river basins mostly in North America, Europe and the Murray-Darling River in Oceania were identified to have a pattern of Knowledge For Environment (KFE). Increases in DM and DI corresponded to decreases in the EN (an inverse indicator on water stress) in these river basins. Generally positive relationships with the SO (median DM = 0.10, DI = 0.74), and counteracting relationships between the DM (-0.02) and DI (0.92) with the EC were also identified. 21 river basins mostly in Asia, Africa and South America had a Knowledge Against Environment (KAE) pattern. These river basins had strong positive

relationships of DM (0.12) and DI (0.93) with EN, SO (DM = 0.28, DI = 0.72), and EC (DM = 0.02, DI = 0.90). Only the DM and DI of the Lake Kinneret were poor predictors to the EN and EC indicators, which was grouped into a separate group identified as "unclear knowledge-society interaction" (Fig. 4a-b, Fig.B3).



25 river basins spreading across North America, Asia, South America and Oceania had a pattern of Knowledge For Resource Availability (KFR). These rivers demonstrated moderately negative relationships of DM (-0.04) and DI (-0.80) with RA (an

inverse indicator on cultivated land). Countering relationships between DM and DI with RU (DM = 0.19, DI = -0.58) and with GC (DM = -0.01, DI = 0.16) were also identified. 16 rivers in Asia or Africa had a Knowledge Against Resources Availability (KAR) pattern, which tended to have strong positive relationships of DM and DI with RA (DM = 0.40, DI = 0.72) and RU (DM = 0.32, DI = 0.63), and counteracting relationships with GC (DM= -0.03, DI = 0.91). The remaining 31 river basins were identified to have "unclear knowledge-policy interaction", mostly in North America. Further, the impacts of DI were generally

stronger and statistically significant whereas much weaker and tended to be insignificant for DM (Fig. 4c-d, Fig. B4).

**(a)**

**(b)**

**(c)**

**(d)**

**Figure 4: (a) The 72 river basins classified based on their linear models between the knowledge structural indicators and the society indicators; and (b) the distributions of the DM and DI coefficients for valid linear models. (c) The 72 river basins classified based on their linear models between the knowledge structural indicators and the policy indicators; and (d) the distributions of the DM and**
**DI coefficients for valid linear models. Dots in the boxplots indicate individual DM and DI coefficients in the linear models, the box**





**boundaries indicate the 25th and 75th percentiles, the centre line indicates the median value, and the whiskers indicate the 1.5 times the interquartile range.**

### 4.4 Optimizing the knowledge structures for improved society and policy

Finally, we identified the DM and DI values that represent the optimal knowledge structures for river basins with each of the
KFE, KAE, KFR, and KAR patterns (see Supplementary Information C for more details). River basins with integrated knowledge structures tended to have KFE (83% of rivers with integrated structures) and KFR (50%) patterns. The issue-driven river basins tended to have KFE (61%) and KAR (38%) patterns, whereas the discipline-driven river basins were dominated by the KFE (94%) and unclear knowledge-policy (75%) patterns. River basins with fragmented knowledge structures were prone to the KAE (36%) and KAR (48%) patterns (Fig. 5a).

For transformation of the knowledge structures of the KFE river basins, an integrated knowledge structure (DM = 1, DI = 1) should be targeted, which maximizes the SO (normalized value = 1) and EC (0.94) indicators while minimizing the negative EN indicators (0.13) (Fig. 5b). On the other hand, there existed trade-offs for the KAE river basins. While the integrated knowledge structure could maximize the SC (1), EN (1), and EC (0.93), a fragmented knowledge structure is optimal to minimize the negative  EN impacts (0.13) but reduced positive SC (0.33) and EC (0.13) impacts (Fig. 5c).

For river basins with the KFR pattern, an integrated structure is optimal to minimize the negative RA (0.14) and maintain balanced RU (0.41) and GC (0.43) (Fig. 5d). For the river basins with the KAR patterns, an integrated knowledge structure could maximize all RA (1), RU (1), and GC (0.89). A fragmented knowledge structures is optimal to minimize the negative RA (0.32), yet traded off with low RU (0.36) and GC (0.10) (Fig. 5e).







**Figure 5: (a) The current knowledge structure, patterns of knowledge impacts on society and policy, and the optimized knowledge structures for the 72 river basins. The optimized society indicators for (b) Knowledge For Environment (KFE) pattern, and (c) Knowledge Against Environment (KAE) pattern; and the optimized policy indicators for (d) Knowledge For Resource Availability (KFR) pattern, and (e) Knowledge Against Resource Availability (KAR) pattern.**

## 5 Discussions and conclusions

Our findings shed light on navigating future knowledge transformation. First, current knowledge structures in global river basins were driven by increasing the Degree of Issue-connectivity (DI) with low and ungrown Degree of Multidisciplinarity



(DM) (Fig. 2). While both integrated and issue-driven knowledge structures have similar optimized societal and policy impacts (Fig. C1), addressing many sustainability issues still requires engaging a broad spectrum of scientific disciplines in the long term. We identified that even for river basins with discipline-driven knowledge structures, the low values of DM and
concentrated interactions among biophysical disciplines imply that multidisciplinary research for most river basins cross the world are still at early stages of development. Therefore, improving multidisciplinary research in particular drawing knowledge from social sciences is the key for transforming the current knowledge systems, especially for those 35% river basins with fragmented knowledge structures.

Second, river basins with the Knowledge Against Environment and the Knowledge Against Resource Availability patterns are
considered less desirable, as optimizing the current knowledge structure to reduce the negative environmental impacts or improving resource availabilities would be traded off with socio-economic development and governance capacities (Fig. 5). Rivers with fragmented knowledge structures comprising 35% of the river basins studied, mostly in Asia, Africa, and South America were most prone to these impact patterns. Additionally, the impacts of DM tended to be statistically insignificant with both society and policy indicators for these rivers (Fig. B3-B4), coupling with the unequal economic development, deteriorated
environment, and depletion of natural resources (Fig. 3). This implies that the effects of multidisciplinary research are not sufficiently transferred to solve the complex sustainability issues. The knowledge systems in these rivers should be reconfigured by strengthening and diversifying environmental and resource protection-driven research.

Third, over 90% of the river basins had knowledge structures that strongly linked to the society indicators but only 57% rivers to the policy, which indirectly explains why water governance (policy) are at crisis globally (Fig. 4). While it has been widely
recognized that policy and practice decisions are informed by diverse values and multiple sources of knowledge, and are shaped by power dynamics beyond the direct influence of research activities (Hakkarainen et al., 2020; Pitt et al., 2018; Posner and Cvitanovic, 2019), we propose to develop "boundary spanning organizations" as a potential solution (Edwards and Meagher, 2020) which can not only bridge disciplinary silos for natural and social scientists, but more importantly to coordinate scientists with local stakeholders and communities with different levels of management powers. Additionally, although beyond
the scope of this study, we recognize the interactions between society and policy. In particular, the SO in society indicators and the RU in policy indicators were most strongly positively correlated ($r = 0.81$, p < 0.05) (Fig. B5).

Finally, about 15% of the river basins studied in America, Europe and Oceania (e.g., the Amazon River, the Colorado River, the Danube River, the Murray-Darling Basin) with integrated knowledge structures demonstrated more balanced impacts on society and policy (Fig. 5). They provide good examples for other river basins, particularly in Asia, Africa and South America
in achieving a holistic integration of science, society and policy.

The limitations in this study and future research directions are also recognized. Only scientific journal publications on the WoS were studied. Classifications of disciplines in this study were conducted based on journal assignments, while boundaries between disciplines have been increasingly blurred when used in the context of research evaluation. The selection of the indicators to represent the society and policy was also bounded by the temporal and spatial data availability. Finally, further
research efforts should be made to reveal the mechanisms behind knowledge structures and the societal and political impacts.





To conclude, this study developed a systemic framework to evaluate the impacts of science on society and policy at a global river basin scale. Rather than using input or output-based knowledge proxies, it directly measured the knowledge structure by using network-based dimensions: Degree of Multidisciplinarity and Degree of Issue-connectivity, which recognizes the diversity and complexity of sustainability issues in the Anthropocene. By determining the structural configurations suitable

for specific outcomes in society and policy, tailored knowledge transformation strategies can be formulated. It also enables comparisons and cumulative learning across cases.

**Appendix A. List of scientific disciplines and issues in the global river basin knowledge network**

Table A1 and A2 summarizes the total number of connections and corresponding percentages for the disciplines and issues identified in the global river basin knowledge network.

**Table A1. Disciplines in the knowledge network**

| Disciplines | No. of connection | Percentage to total connection |
|---|---|---|
| Environmental Sciences | 30398 | 19.63% |
| Water Resources | 16581 | 10.71% |
| Geosciences, Multidisciplinary | 11578 | 7.48% |
| Marine & Freshwater Biology | 11428 | 7.38% |
| Ecology | 10703 | 6.91% |
| Engineering, Environmental | 6045 | 3.90% |
| Limnology | 5606 | 3.62% |
| Engineering, Civil | 4826 | 3.12% |
| Meteorology & Atmospheric Sciences | 4809 | 3.10% |
| Geography, Physical | 4034 | 2.60% |
| Fisheries | 3260 | 2.10% |
| Oceanography | 3187 | 2.06% |
| Biodiversity Conservation | 2947 | 1.90% |
| Environmental Studies | 2622 | 1.69% |
| Toxicology | 2199 | 1.42% |
| Zoology | 1760 | 1.14% |
| Agronomy | 1733 | 1.12% |
| Soil Science | 1713 | 1.11% |
| Multidisciplinary Sciences | 1542 | 1.00% |





| | | |
|---|---|---|
| Public, Environmental & Occupational Health | 1366 | 0.88% |
| Geochemistry & Geophysics | 1355 | 0.87% |
| Plant Sciences | 1279 | 0.83% |
| Geography | 1231 | 0.79% |
| Green & sustainable science & technology | 883 | 0.57% |
| Evolutionary Biology | 876 | 0.57% |
| Remote Sensing | 862 | 0.56% |
| Economics | 825 | 0.53% |
| Forestry | 825 | 0.53% |
| Genetics & Heredity | 781 | 0.50% |
| Agriculture, Multidisciplinary | 773 | 0.50% |
| Biochemistry & Molecular Biology | 740 | 0.48% |
| Chemistry, Analytical | 641 | 0.41% |
| Engineering, Chemical | 615 | 0.40% |
| Energy & Fuels | 613 | 0.40% |
| Agricultural Engineering | 579 | 0.37% |
| Geology | 542 | 0.35% |
| Biology | 532 | 0.34% |
| Biotechnology & Applied Microbiology | 529 | 0.34% |
| Anthropology | 526 | 0.34% |
| Microbiology | 522 | 0.34% |
| Imaging Science & Photographic Technology | 521 | 0.34% |
| Urban Studies | 474 | 0.31% |
| Ornithology | 464 | 0.30% |
| Planning & Development | 435 | 0.28% |
| Chemistry, Multidisciplinary | 413 | 0.27% |
| Computer Science, Interdisciplinary Applications | 413 | 0.27% |
| Entomology | 398 | 0.26% |
| Engineering, Geological | 366 | 0.24% |
| Paleontology | 357 | 0.23% |





| | | |
|---|---|---|
| Veterinary Sciences | 308 | 0.20% |
| Food Science & Technology | 285 | 0.18% |
| Archaeology | 279 | 0.18% |
| Statistics & Probability | 256 | 0.17% |
| Sociology | 251 | 0.16% |
| Engineering, Mechanical | 236 | 0.15% |
| Materials Science, Multidisciplinary | 224 | 0.14% |
| Area Studies | 215 | 0.14% |
| Engineering, Multidisciplinary | 213 | 0.14% |
| Horticulture | 212 | 0.14% |
| Nuclear Science & Technology | 211 | 0.14% |
| Law | 210 | 0.14% |
| Political Science | 209 | 0.13% |
| Engineering, Ocean | 193 | 0.12% |
| Parasitology | 193 | 0.12% |
| Social Sciences, Interdisciplinary | 184 | 0.12% |
| Mathematics, Interdisciplinary Applications | 180 | 0.12% |
| Transportation | 176 | 0.11% |
| Operations Research & Management Science | 170 | 0.11% |
| Agricultural Economics & Policy | 165 | 0.11% |
| Biochemical Research Methods | 154 | 0.10% |
| Engineering, Electrical & Electronic | 152 | 0.10% |
| History | 147 | 0.09% |
| International Relations | 142 | 0.09% |
| Chemistry, Inorganic & Nuclear | 141 | 0.09% |
| Mechanics | 136 | 0.09% |
| Management | 131 | 0.08% |
| Infectious Diseases | 128 | 0.08% |
| Tropical Medicine | 127 | 0.08% |
| Public Administration | 127 | 0.08% |
| Construction & Building Technology | 122 | 0.08% |



| | | |
|---|---|---|
| Chemistry, Applied | 119 | 0.08% |
| Agriculture, Dairy & Animal Science | 117 | 0.08% |
| Transportation Science & Technology | 109 | 0.07% |
| Business | 104 | 0.07% |
| Social Sciences, Mathematical Methods | 96 | 0.06% |
| Thermodynamics | 92 | 0.06% |
| Chemistry, Physical | 90 | 0.06% |
| History & Philosophy Of Science | 87 | 0.06% |
| Radiology, Nuclear Medicine & Medical Imaging | 85 | 0.05% |
| Physiology | 84 | 0.05% |
| Instruments & Instrumentation | 82 | 0.05% |
| Information Science & Library Science | 80 | 0.05% |
| Computer Science, Information Systems | 80 | 0.05% |
| Hospitality, Leisure, Sport & Tourism | 73 | 0.05% |
| Engineering, Industrial | 73 | 0.05% |
| Endocrinology & Metabolism | 72 | 0.05% |
| Immunology | 71 | 0.05% |
| Mining & Mineral Processing | 65 | 0.04% |
| History Of Social Sciences | 64 | 0.04% |
| Computer Science, Artificial Intelligence | 60 | 0.04% |
| Pharmacology & Pharmacy | 58 | 0.04% |
| Mineralogy | 57 | 0.04% |
| Electrochemistry | 57 | 0.04% |
| Physics, Multidisciplinary | 55 | 0.04% |
| Behavioral Sciences | 53 | 0.03% |
| Spectroscopy | 50 | 0.03% |
| Engineering, Marine | 48 | 0.03% |
| Medicine, General & Internal | 44 | 0.03% |
| Metallurgy & Metallurgical Engineering | 43 | 0.03% |
| Demography | 42 | 0.03% |
| Mathematics, Applied | 41 | 0.03% |





| | | |
|---|---|---|
| Nutrition & Dietetics | 38 | 0.02% |
| Engineering, Petroleum | 37 | 0.02% |
| Health Care Sciences & Services | 37 | 0.02% |
| Architecture | 37 | 0.02% |
| Materials Science, Paper & Wood | 34 | 0.02% |
| Education & Educational Research | 33 | 0.02% |
| Social Sciences, Biomedical | 32 | 0.02% |
| Health Policy & Services | 32 | 0.02% |
| Medicine, Research & Experimental | 31 | 0.02% |
| Engineering, Manufacturing | 30 | 0.02% |
| Physics, Mathematical | 28 | 0.02% |
| Physics, Fluids & Plasmas | 26 | 0.02% |
| Computer Science, Software Engineering | 25 | 0.02% |
| Physics, Applied | 25 | 0.02% |
| Communication | 25 | 0.02% |
| Polymer Science | 24 | 0.02% |
| Biophysics | 24 | 0.02% |
| Medicine, Legal | 23 | 0.01% |
| Virology | 23 | 0.01% |
| Automation & Control Systems | 23 | 0.01% |
| Computer Science, Theory & Methods | 22 | 0.01% |
| Developmental Biology | 22 | 0.01% |
| Women's Studies | 21 | 0.01% |
| Materials Science, Characterization & Testing | 20 | 0.01% |
| Cultural Studies | 20 | 0.01% |
| Physics, Nuclear | 19 | 0.01% |
| Neurosciences | 19 | 0.01% |
| Industrial Relations & Labor | 18 | 0.01% |
| Pathology | 18 | 0.01% |
| Cell Biology | 18 | 0.01% |
| Psychology, Multidisciplinary | 18 | 0.01% |





| | | |
|---|---|---|
| Ethics | 17 | 0.01% |
| Nanoscience & Nanotechnology | 16 | 0.01% |
| Pediatrics | 15 | 0.01% |
| Mathematical & Computational Biology | 15 | 0.01% |
| Chemistry, Organic | 14 | 0.01% |
| Physics, Atomic, Molecular & Chemical | 14 | 0.01% |
| Astronomy & Astrophysics | 12 | 0.01% |
| Linguistics | 12 | 0.01% |
| Ethnic Studies | 11 | 0.01% |
| Psychiatry | 11 | 0.01% |
| Education, Scientific Disciplines | 10 | 0.01% |
| Optics | 10 | 0.01% |
| Reproductive Biology | 10 | 0.01% |
| Sport Sciences | 10 | 0.01% |
| Language & Linguistics | 10 | 0.01% |
| Social Issues | 9 | 0.01% |
| Mycology | 9 | 0.01% |
| Chemistry, Medicinal | 9 | 0.01% |
| Dentistry, Oral Surgery & Medicine | 8 | 0.01% |
| Art | 8 | 0.01% |
| Physics, Condensed Matter | 8 | 0.01% |
| Telecommunications | 8 | 0.01% |
| Acoustics | 8 | 0.01% |
| Materials Science, Ceramics | 7 | 0.00% |
| Oncology | 7 | 0.00% |
| Psychology, Clinical | 6 | 0.00% |
| Respiratory System | 6 | 0.00% |
| Audiology & Speech-Language Pathology | 6 | 0.00% |
| Otorhinolaryngology | 6 | 0.00% |
| Criminology & Penology | 6 | 0.00% |
| Substance Abuse | 6 | 0.00% |
| Nursing | 6 | 0.00% |



| | | |
|---|---|---|
| Psychology, Educational | 6 | 0.00% |
| Anesthesiology | 5 | 0.00% |
| Computer Science, Hardware & Architecture | 5 | 0.00% |
| Allergy | 5 | 0.00% |
| Ergonomics | 5 | 0.00% |
| Family Studies | 5 | 0.00% |
| Asian Studies | 5 | 0.00% |
| Urology & Nephrology | 5 | 0.00% |
| Business, Finance | 5 | 0.00% |
| Obstetrics & Gynecology | 4 | 0.00% |
| Surgery | 4 | 0.00% |
| Ophthalmology | 4 | 0.00% |
| Clinical Neurology | 4 | 0.00% |
| Psychology, Developmental | 4 | 0.00% |
| Humanities, Multidisciplinary | 4 | 0.00% |
| Psychology | 4 | 0.00% |
| Primary Health Care | 4 | 0.00% |
| Physics, Particles & Fields | 3 | 0.00% |
| Anatomy & Morphology | 3 | 0.00% |
| Geriatrics & Gerontology | 3 | 0.00% |
| Film, Radio, Television | 3 | 0.00% |
| Materials Science, Textiles | 3 | 0.00% |
| Integrative & Complementary Medicine | 3 | 0.00% |
| Psychology, Social | 3 | 0.00% |
| Crystallography | 2 | 0.00% |
| Microscopy | 2 | 0.00% |
| Critical Care Medicine | 2 | 0.00% |
| Social Work | 2 | 0.00% |
| Psychology, Applied | 2 | 0.00% |
| Materials Science, Biomaterials | 2 | 0.00% |
| Medical Ethics | 2 | 0.00% |





| | | |
|---|---|---|
| Emergency Medicine | 2 | 0.00% |
| Peripheral Vascular Disease | 1 | 0.00% |
| Mathematics | 1 | 0.00% |
| Computer Science, Cybernetics | 1 | 0.00% |
| Religion-dis | 1 | 0.00% |
| Gerontology | 1 | 0.00% |
| Gastroenterology & Hepatology | 1 | 0.00% |
| Logic | 1 | 0.00% |
| Engineering, Biomedical | 1 | 0.00% |
| Psychology, Experimental | 1 | 0.00% |

**Table A2. Issues in the knowledge network**

| Issues | No. of connection | Percentage to total connection |
|---|---|---|
| Ecological degradation and restoration | 25913 | 16.50% |
| Pollution and treatment | 20914 | 13.31% |
| Management and control | 9328 | 5.94% |
| Agriculture and irrigation | 7314 | 4.66% |
| Flood and drought and their mitigation | 7195 | 4.58% |
| Erosion and sedimentation | 5894 | 3.75% |
| Climate change | 4985 | 3.17% |
| Water scarcity and availability | 4474 | 2.85% |
| Population | 3474 | 2.21% |
| Risk and impact assessment | 3337 | 2.12% |
| Other hazard | 2917 | 1.86% |
| Salinity and alkalinity | 2867 | 1.83% |
| Urban issue | 2610 | 1.66% |
| Other climatic extreme | 2447 | 1.56% |
| Land use change | 2411 | 1.53% |
| Hydropower | 2246 | 1.43% |
| General economic development | 2217 | 1.41% |
| Pesticide and fertilisation | 2037 | 1.30% |
| Construction | 1893 | 1.21% |





| | | |
|---|---|---|
| Plan and strategy | 1859 | 1.18% |
| Human activity | 1827 | 1.16% |
| Hydrological change | 1815 | 1.16% |
| Transportation | 1715 | 1.09% |
| Regulation | 1705 | 1.09% |
| Energy | 1559 | 0.99% |
| Aquaculture and fishery | 1524 | 0.97% |
| Value | 1441 | 0.92% |
| Population migration | 1286 | 0.82% |
| History | 1193 | 0.76% |
| Policy | 1097 | 0.70% |
| Public health | 1047 | 0.67% |
| Government | 992 | 0.63% |
| Vegetation and desertification | 981 | 0.62% |
| Conflict | 923 | 0.59% |
| Biodiversity | 884 | 0.56% |
| Decision making | 880 | 0.56% |
| Drinking water and salinisation | 875 | 0.56% |
| Forecasting | 842 | 0.54% |
| Carbon emission and sequestration | 813 | 0.52% |
| General societal issue | 797 | 0.51% |
| Behaviour | 783 | 0.50% |
| Monitoring | 751 | 0.48% |
| Trading and entitlement | 748 | 0.48% |
| Sustainability | 715 | 0.46% |
| Industry | 678 | 0.43% |
| Governance | 658 | 0.42% |
| Mapping and tool | 655 | 0.42% |
| Sea surface change | 648 | 0.41% |
| Law | 627 | 0.40% |
| Operation | 595 | 0.38% |
| Tourism and recreation | 578 | 0.37% |





| | | |
|---|---|---|
| Precipitation change | 528 | 0.34% |
| Collaboration | 488 | 0.31% |
| Food security | 487 | 0.31% |
| Transition | 455 | 0.29% |
| Mining | 430 | 0.27% |
| Rural issue | 427 | 0.27% |
| Other natural resources | 391 | 0.25% |
| Technology development | 360 | 0.23% |
| Knowledge and capacity | 359 | 0.23% |
| Standard | 325 | 0.21% |
| Geological change | 298 | 0.19% |
| Pharmacy | 293 | 0.19% |
| Politics | 286 | 0.18% |
| Socio-ecological | 279 | 0.18% |
| Stakeholder engagement | 273 | 0.17% |
| Social event | 273 | 0.17% |
| Inequality | 250 | 0.16% |
| Temperature rise | 234 | 0.15% |
| Education and training | 218 | 0.14% |
| Greenhouse gas increase | 211 | 0.13% |
| Subsidy | 209 | 0.13% |
| Class and ethnicity | 206 | 0.13% |
| Gender | 204 | 0.13% |
| Globalisation | 202 | 0.13% |
| Human health | 199 | 0.13% |
| Prospect and vision | 192 | 0.12% |
| Emergency | 178 | 0.11% |
| Forestry | 133 | 0.08% |
| Textile and paper mill | 124 | 0.08% |
| Media and communication | 103 | 0.07% |
| Public affairs | 97 | 0.06% |
| Climate change mitigation and adaptation | 87 | 0.06% |





| | | |
|---|---|---|
| Relation | 81 | 0.05% |
| Civilisation | 55 | 0.04% |
| Permit | 40 | 0.03% |
| Employment | 31 | 0.02% |
| Citizenship | 25 | 0.02% |
| Science-policy | 24 | 0.02% |
| Literature and language | 21 | 0.01% |
| Power | 12 | 0.01% |
| Art | 9 | 0.01% |
| Crime | 8 | 0.01% |
| Religion | 2 | 0.00% |





**Appendix B. Additional statistical details**

Figure B1-B5 provides additional statistical details on the results, as explained in corresponding figure captions.

**Figure B1. (a) The temporal trends (Sen's slope) and (b) the absolute values of the social indicators; (c) the temporal trends (Sen's slope) and (d) the absolute values of the economic indicators; (e) the temporal trends (Sen's slope) and (f) the absolute values of the environmental indicators for the 72 river basins. Dots in the boxplots indicate individual DM**



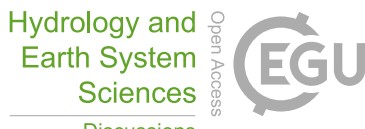

**and DI values, the box boundaries indicate the 25ᵗʰ and 75ᵗʰ percentiles, the centre line indicates median values, and the whiskers indicate the 1.5 times of the interquartile range.**

**Figure B2. (a) The temporal trends (Sen's slope) and (b) the absolute values of the resource availability indicators; (c) the temporal trends (Sen's slope) and (d) the absolute values of the resource utilization indicators; (e) the temporal**

**trends (Sen's slope) and (f) the absolute values of the governance capacity indicators for the 72 river basins. Dots in the**





boxplots indicate individual DM and DI values, the box boundaries indicate the 25$^{th}$ and 75$^{th}$ percentiles, the centre line indicates median values, and the whiskers indicate the 1.5 times of the interquartile range.

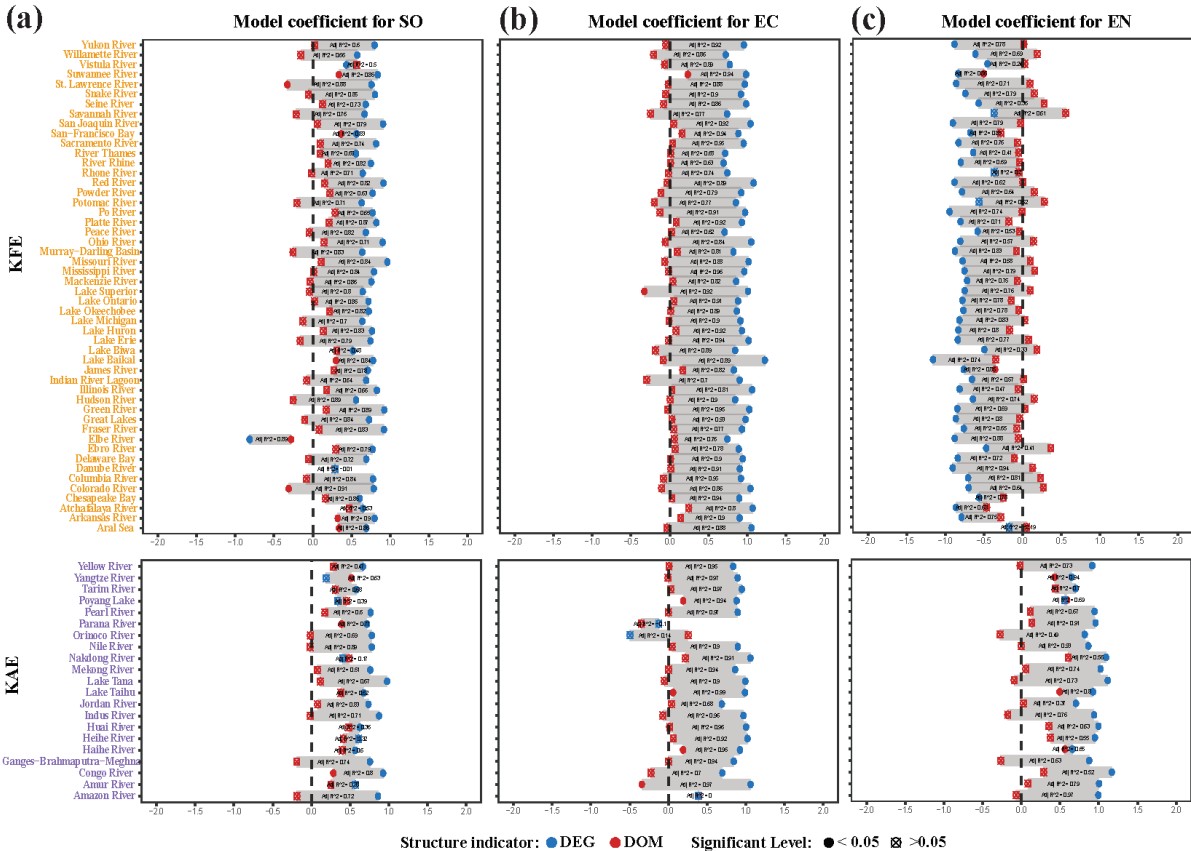

**Figure B3. The model coefficients for each river basin's linear models between the structural indicators and the (a) SO indicators, (b) the EC indicators, and (c) the EN indicators.**





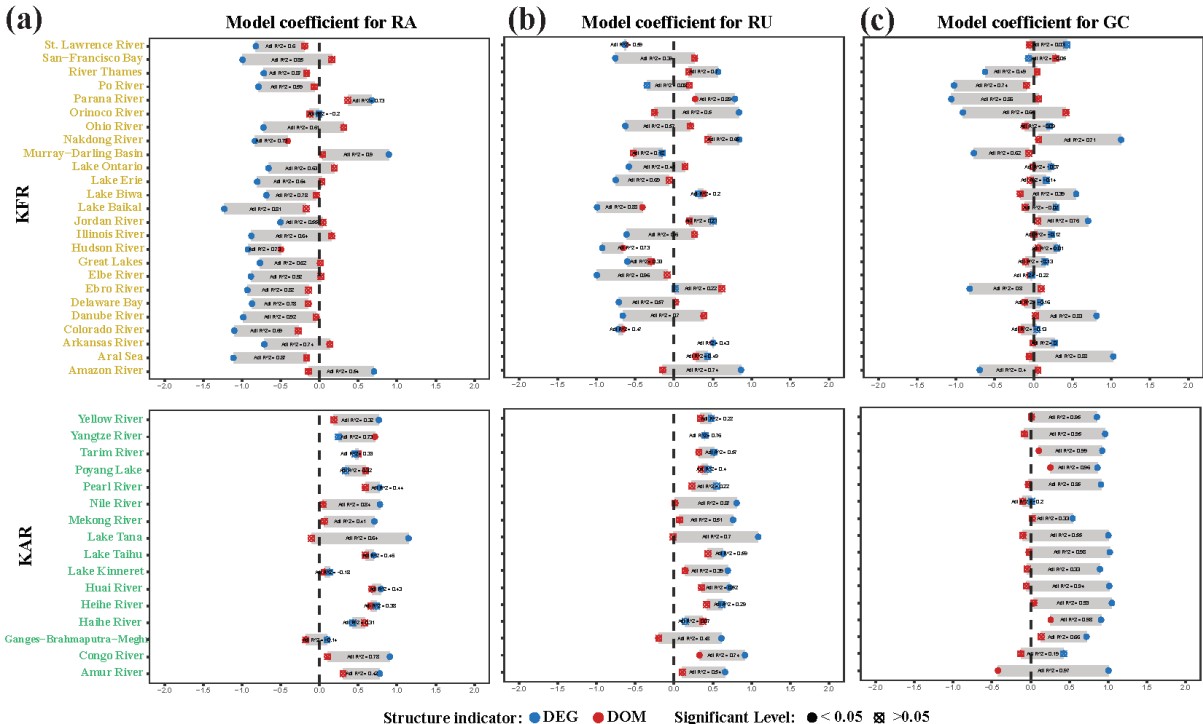

**Figure B4. The model coefficients for each river basin's linear models between the structural indicators and the (a) RA indicators, (b) the RU indicators, and (c) the GC indicators.**





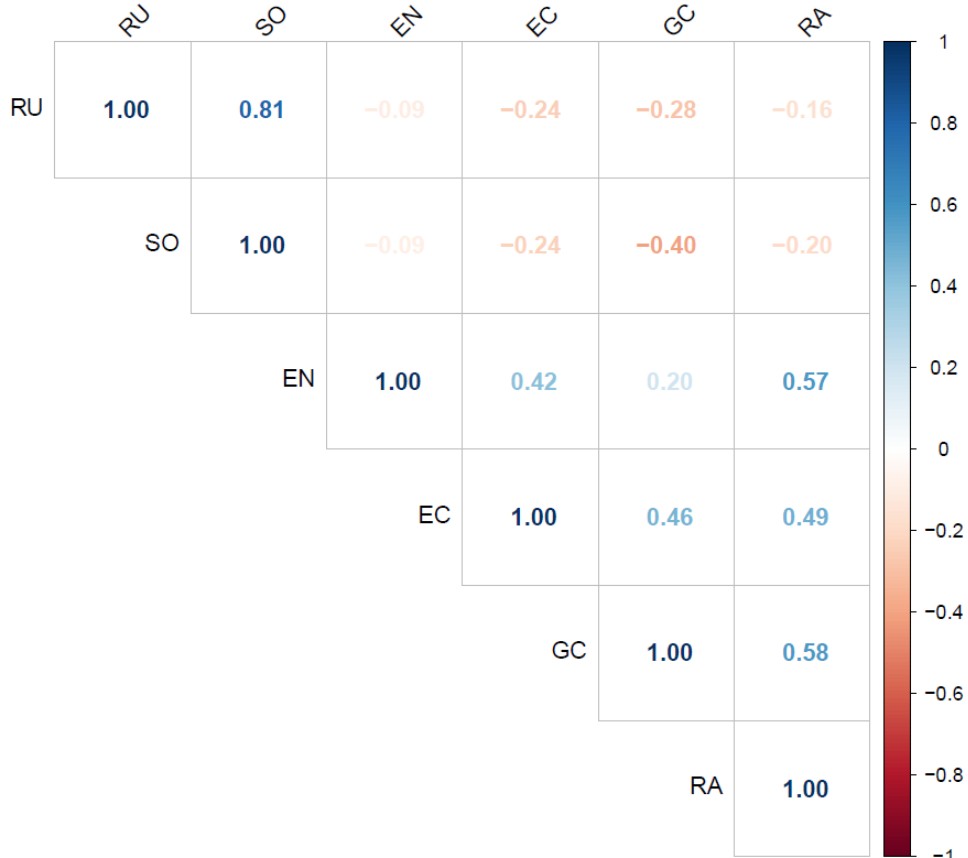


**Figure B5. The Pearson correlations among the society system and policy system indicators, indicators are ordered based on hierarchical clustering using Ward's Distance.**





**Appendix C. Additional methods and results on optimizing the knowledge structures for improved society and policy**

The clustering analysis of river basins based on the regression models for the society indicators (i.e., SO, EC, and EN) resulted
in three knowledge-society interaction patterns for the 72 river basins: the Knowledge For Environment (KFE), the Knowledge
Against Environment (KAE), and the unclear knowledge-society interaction patterns. Similarly, the regression models for the
policy indicators (i.e., RA, RU, and GC) resulted in three knowledge-policy interaction patterns: the Knowledge For Resource
availability (KFR), the Knowledge Against Resource availability (KAR), and the unclear knowledge-policy interaction
patterns. This means that each of the 72 river basins have one knowledge-society interaction pattern and one knowledge-policy
interaction pattern.

To identify the knowledge structures (i.e., DM and DI) that optimize the society indicators, we first removed the rivers with
unclear knowledge-society interaction pattern (n=1), and then calculated the average regression coefficients for river basins
under the KFE and KAE patterns, respectively. Similarly to identify the DM and DI values for optimized policy indicators,
rivers with unclear knowledge-policy interaction patterns were removed (n=31), and the average regression coefficients for
each of the KFR and KAR patterns were calculated. This resulted in 12 regression relationships (two for each of SO, EC, EN,
RA, RU, and GC), as summarized in Table C1.

**Table C1. Knowledge-impact relationships used as objective functions for optimization**

| Society indicator | Knowledge-society pattern: KFE (n = 50) | Knowledge-society pattern: KAE (n = 21) |
|---|---|---|
| SO | $= 0.075 \times DM'_{opt} + 0.692 \times DI'_{opt} + 0.282$ | $= 0.223 \times DM'_{opt} + 0.674 \times DI'_{opt} + 0.331$ |
| EC | $= -0.016 \times DM'_{opt} + 0.919 \times DI'_{opt} + 0.034$ | $= 0.022 \times DM'_{opt} + 0.774 \times DI'_{opt} + 0.133$ |
| EN | $= -0.002 \times DM'_{opt} - 0.734 \times DI'_{opt} + 0.863$ | $= 0.175 \times DM'_{opt} + 0.899 \times DI'_{opt} + 0.133$ |
| Policy indicator | Knowledge-policy pattern: KFR (n = 25) | Knowledge-policy pattern: KAR (n = 16) |
| RA | $= -0.045 \times DM'_{opt} - 0.626 \times DI'_{opt} + 0.789$ | $= 0.338 \times DM'_{opt} + 0.613 \times DI'_{opt} + 0.315$ |
| RU | $= 0.025 \times DM'_{opt} - 0.177 \times DI'_{opt} + 0.565$ | $= 0.230 \times DM'_{opt} + 0.627 \times DI'_{opt} + 0.362$ |
| GC | $= 0.001 \times DM'_{opt} + 0.024 \times DI'_{opt} + 0.402$ | $= -0.010 \times DM'_{opt} + 0.819 \times DI'_{opt} + 0.085$ |

For each of the KFE, KAE, KFR, and KAR pattern, these relationships were used as objective functions for multi-objective
optimizations using a NSGA-II genetic algorithm (Deb et al., 2002; Coello coello et al., 2020) to identify the optimal DM and
DI values ($DM'_{opt}$ , $DI'_{opt}$). 100 pairs of potential DM and DI values were randomly generated initially and modelled over 1000
iterations to search for the optimum values that achieve the objectives as outlined in Table 2.

The global Pareto optimality for each pattern were identified when the Pareto Front = 1, which indicated the set of effective
solutions that were at least as good as other possible solutions for each objective and strictly better for at least one objective
(Halffmann et al., 2022). Set of optimal DM and DI values that resulted in society and policy indicators for each pattern were
identified, as shown in Figure C1.





The dark coloured lines highlight the boundary values for the SO, EC, EN, RA, RU, and GC indicators, and their corresponding DM and DI values that were selected as the optimal solutions discussed in the main text. The light coloured lines represent the other possible values on the Pareto Front.

It should also be noted that as we conducted optimizations based on the average coefficients in the linear models, these exact optimal DM and DI values were not directly related to any specific rivers in each knowledge-impact pattern group. Therefore, we referred to the corresponding knowledge structures (i.e., integrated, issue-driven, discipline-driven, and fragmented) that these structural values represented as the optimal knowledge structures that achieved the society and policy objectives.

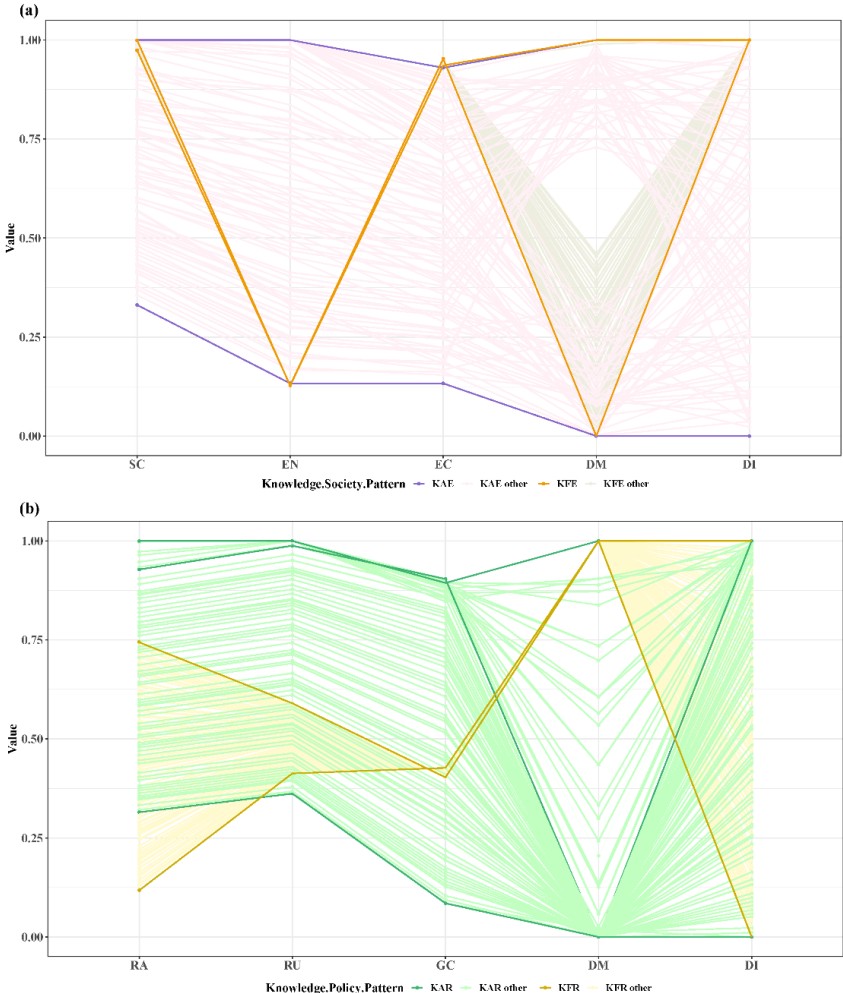

**Figure C1. The pareto front values for the (a) society and (b) policy indicators, and the corresponding DM and DI**
**values.**



**Data and code availability**

Data pertaining to this work is available publicly as cited in the manuscript, and codes used to analyse the data is deposited in https://github.com/SLWU423/Code-for-global-river-basin-science-policy-society-impact.

**Author contribution**

S. Wu contributed to conceptualization, data curation, methodology, data analysis, writing the original draft, reviewing and editing the manuscript; Y. Wei contributed to conceptualization, methodology, data validation, reviewing and editing the manuscript.

**Competing interests**

At least one of the (co-)authors is a member of the editorial board of HESS.

**Acknowledgement**

This study is supported by the Australian Research Council Special Research Initiative [SR200200186], and the University of Queensland Research Stimulus (UQ RS) Fellowship.

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
