# Peer review of "Impacts of science on society and policy in major river basins globally"

_Hydrology and Earth System Sciences, 2024_

## Author Comment (AC1)

**RC1: Comment on hess-2024-72**

The manuscript systematically elaborates on the knowledge system of key river. Among them, the key processes and algorithms has its further selection principles or unique applicability, and there is a detailed introduction. This is very important in the evaluation of knowledge system services and is more conducive to comparison between different studies. However, after overall review, there are still many doubts regarding the following:

Overall Comment:

(1) Formula Standardization

There are descriptive words in formulas 1, 2, and 3 in the manuscript. Please explain the variables and their meanings in a more explicit way (e.g. "where, x is ... " in Eq.3), and use a more explicit variable calculation form to present the "median" (Eq.3).

Thank you for your comments. We will refine and add further explanations for Eq.1-3 as follows:

For any discipline-issue network i:

$$DM_i = \frac{2C_d}{n(n-1)} \qquad \text{(Eq.1)}$$

where $DM_i$ is the Degree of Multidisciplinarity value of a discipline-issue network i, $C_d$ is the total number of existing connections between any issue and discipline d in the network, and n is the total number of d in the network.

For any issue network i:

$$DI_i = \frac{\sum_n C_m}{n} \qquad \text{(Eq.2)}$$

where $DI_i$ is the Degree of Issue-connectivity of an issue network i, $C_m$ is the number of adjacent connections to any specific issue m, and n is the total number of m in the network.

The Sen's slopes (Sen, 1968) were then used to measure the magnitudes of the trends as Eq.3:

$$d_{Sen} = \tilde{d}\left(\frac{x_j - x_i}{j - i}\right) \text{ for } 1 \le i < j \le n \qquad \text{(Eq.3)}$$

where $\tilde{d}$ is the median value separating the higher 50% from the lower 50% of the indicator value x in the time series, $i$ and $j$ are adjacent time points, and n is the total number of time points.

(2) Structural Issues

Can the representation of the framework in Section 2 be considered as a preface to the Methods section? Among them, the density of the discipline-issue network and the calculation method for the degree centrality of the issue network are all in the Section 3. It becomes clearer whether they can be merged.

Thank you for your comments. We will integrate Section 2 into the Method section as a new Section 2.1, and consolidate the descriptions of the framework and relevant calculations about the discipline-issue network and the issue network in the new Section 2.1.

(3) Comprehensive Knowledge Structure

The selection of 72 river basins is mostly typical of river systems in various continents, and is also significantly influenced by human activities. And the information must also be relatively detailed, which is a necessary foundation for this research method. However, can the representativeness of social and policy analysis be highlighted based on existing analysis results? ("Abstract: ...Evaluating these structural characteristics against 6 impact indicators on society and policy, over 90% of rivers were found to had knowledge structures that strongly linked to societal impacts whereas only 57% were to the policy...") After all, the title mentions global river basins, but currently the intuitive feeling is to search for conclusions in large rivers influenced by humans, which always feels somewhat inappropriate. Please take above concern into consideration.

Additionally, why are there missing rivers in the North Asian region, such as the Ob River and Yenisei River basins? Will the North Asian rivers, which are relatively low in human activity, affect the relevant conclusions on policy and social impact in the abstract?

Thank you for your comments. The 72 river basins were selected based on those receiving the highest numbers of publications in the WoS database. We chose peer-reviewed publications in the WoS as our data source as it provides consistent, systematic documentation of scientific knowledge development across a broad range of disciplines for a long timeframe. At least one river basin in each of the continents was included for the spatial representativeness of this study.

However, we agree that there is a potential bias towards large river basins with societal and natural significance to be studied, and some rivers may not be included due to comparatively fewer publications in the WoS. For example, the Lake Baikal catchment was studied, which was a major part of the Yenisei River. We will clarify this in the method section and recognize it as a limitation in the discussion section.

In addition, we will change our title as "Impacts of science on society and policy in main river basins in the world" to better reflect the scope of the study.

(4) Support for key conclusions in the manuscript

The following sentence is an explanation of the key conclusions in the abstract ("Abstract: ...over 90% of rivers were found to had knowledge structures that strongly linked to societal impacts whereas only 57% were to the policy"). However, is the $R^2$ the smallest among the 41 basins greater than 0.3, or is the mean of the 41 basins greater than 0.3? The $R^2$ value is indeed a bit low, and the correlation explanation is weak; But it is possible that in such studies, more than 0.3 has already met the interpretive requirements. The manuscript can supplement the general situation of $R^2$ in similar studies and compare the level of 0.3. To enhance the reliability of the conclusions of this article.

" The structural characteristics of the knowledge systems had been strongly linked to the society indicators with over 90% river basins had acceptable regression model fits, but much weaker with the policy indicators as only 41 river basins had two or more linear models that validated the relationships between their knowledge systems and the policy (adjusted $R^2 > 0.3$, statistical significance $p < 0.05$). "

Thank you for your comments. The $R^2$ values in this study were estimated in each regression model for each river basin, and any models with $R^2$ values smaller than 0.3 were grouped into the 'unclear knowledge-society' or 'unclear knowledge-policy' pattern.

The threshold of 0.3 was selected based on studies in conventional statistical regressions (Ratner, 2009; Royston, 2007), which identified 0.3 to have "weak" explanation power between the knowledge indicators and society/policy indicators. Similar thresholds between 0.2 and 0.3 have also been found by correlations between knowledge, attitudes, and practices regarding environmental problems (Afroz & Ilham, 2020; Alias, 2019). In general, a recent meta-analysis (Hernanda et al., 2023) indicated an acceptable range for correlation levels to be 0.26 to 0.48 across 23 studies published from 1999 to 2022. We will provide this additional justification in the method section.

(5) Section of "Data and code availability"

(Only representing personal opinions) Compared to conclusive summaries, collecting and organizing information and making accurate judgments in the process will be more important. Can the manuscript be supplemented with information about the data or list of statistically analyzed in the article, in order to facilitate further research development or review during the evaluation process.

Thank you for your comments. We have provided an Appendix document, and will add additional explanations for each section detailing the data information and statistical analysis conducted to support the results in the manuscript. R codes used to generate the results were also commented and deposited in the public repository Github

(https://github.com/SLWU423/Code-for-global-river-basin-science-policy-society-impact) for reproduction of the results and further research development.

**General Comment:**

**(1) Image clarity**

The text resolution in Figure 2-c is not sufficient to see clearly, and there is overlap with the 0-axis. Is the threshold for "low DM" or "high DI" in the manuscript Line 230~235) divided by the 25th and 75th percentiles in box boundaries?

The resolution in the all figures is not clear, especially in the form of coordinate axis subfigures.

Thank you for your comments. We refrained from introducing additional subjective bias to define a specific threshold value for DM and DI, and considered the comparative values of DM and DI among the 72 river basins. Therefore, the low and high DM and DI were determined by their z-scores:

For any river basin k, and any knowledge, societal, and policy indicator x:

$$x'_k = \frac{x_k - \overline{x_k}}{\sigma_k}$$

where $x'_k$ is the z-score of any knowledge, societal and policy indicator of $x_k$ , $\overline{x_k}$ is the mean value, and $\sigma_k$ is the standard deviation.

Therefore, we determined the division between 'low' and 'high' scores by the zero value of z-score. A z-score above zero means that the DM or DI value is above the average value for all rivers, and therefore having a 'high' DM or DI. Similarly, a z-score value below zero will be considered having a 'low' DM or DI. This will be clarified in the framework in the new Section 2.1 and in the Result Section 3.1 in the revised manuscript.

We will also increase the resolutions and fonts for all figures for improved clarity in the revised manuscript.

**(2) Optimization processing of Appendix**

The table in the Appendix only requires quantity, and the proportion of 0.00% is the result of omitted accuracy. The number of columns can be changed to reduce pages (Table A1, Table A2).

Thank you for your comments. We will remove the proportion values and reformat all tables in Appendix A.

**References:**

Afroz, N., & Ilham, Z. (2020). Assessment of Knowledge, Attitude and Practice of University Students towards Sustainable Development Goals (SDGs). *The Journal of Indonesia Sustainable Development Planning*, *1*(1), 31-44. https://doi.org/10.46456/jisdep.v1i1.12

Alias, N. A. (2019). Correlation between knowledge, attitude, and behavior towards river pollution. *International Journal of Modern Trends in Social Sciences*, *2*(9), 31-38.

Hernanda, T., Absori, Azhari, A. F., Wardiono, K., & Arlinwibowo, J. (2023). Relationship Between Knowledge and Affection for the Environment: A Meta-Analysis. *European Journal of Educational Research*, *12*(2), 1071-1084. https://doi.org/10.12973/eu-jer.12.2.1069

Ratner, B. (2009). The correlation coefficient: Its values range between +1/−1, or do they? *Journal of Targeting, Measurement and Analysis for Marketing*, *17*(2), 139-142. https://doi.org/10.1057/jt.2009.5

Royston, P. (2007). Profile Likelihood for Estimation and Confidence Intervals. *The Stata Journal*, *7*(3), 376-387. https://doi.org/10.1177/1536867x0700700305

Sen, P. K. (1968). Estimates of the Regression Coefficient Based on Kendall's Tau. *Journal of the American Statistical Association*, *63*(324), 1379-1389. https://doi.org/10.1080/01621459.1968.10480934

---

## Author Comment (AC2)

**RC2: Comment on hess-2024-72**

This study on the impacts of scientific knowledge development on society and policy within global river basins is both timely and insightful. The framework for measuring knowledge systems through network dimensions of multidisciplinarity and issue-connectivity is commendable. Here are some review comments:

Framework and Methodology: Elaborate on the theoretical underpinnings of your proposed framework and discuss its potential for long-term applicability.

Thank you for your comments. We will consolidate the theoretical underpinning that supports the knowledge network construction in a new Section 2.1. Specifically, the framework was developed based on:

1) The Science of Science (SoS) theory (Zeng et al., 2017) which conceptualised knowledge as a dynamic system consisting of numerous disciplinary knowledge and the issues studied, with complex and co-evolving relationships.
2) The network theory and our previous studies (Fortunato et al., 2018; Wei & Wu, 2022; Wu et al., 2021) which provide a systematic approach to quantifying such complex interactions as the structure of the knowledge system. The structure of the knowledge system then determined the functionality (i.e., impacts) of the overall knowledge system (Coccia, 2020; Huttenhower et al., 2012; Sayles & Baggio, 2017; Von Bertalanffy, 1968).

We will add an additional section in the Discussion section elaborating the implications of the framework, which will be contributing to future knowledge transformation for more sustainable river basin development by:

1) providing a method to explicitly measure the structure of a knowledge system as a discipline-issue network, which guides future knowledge development by identifying explicitly where and what to change or connect between disciplinary knowledge and issues at hand, therefore assisting in more suitable, more precise, and more predictable knowledge development in the future.
2) linking the structural configurations of knowledge systems with developments in the society and policy, thus contributing to better evaluation of research outcomes and action-oriented research for specifying the "credible, legitimate, and relevant" criteria in good governance (Cash et al., 2003; Kim, 2019).
3) enabling comparisons of knowledge development for river basins with varying management issues of focuses and contexts, thus enabling the design of tailored management strategies and co-learning according to different patterns of connections among river basin knowledge, society, and policy development.

Data and Analysis: Consider the inclusion of additional data sources beyond Web of Science, such as conference papers or government reports, to enhance the study's comprehensiveness.

Thank you for your comments. This study focused on the science-driven knowledge development, by using peer-reviewed articles indexed in the Web of Science (WoS) database. The WoS database was chosen because it provides consistent, systematic documentation of knowledge development across a broad range of disciplines for a long timeframe. However, we do acknowledge that use of additional data including conference paper, and government reports also contributes to the river basin knowledge development, which tends to focus on the practice-driven knowledge. This will be recognized as a limitation in the discussion section.

Address the potential variability in keyword extraction and clustering across different languages and regions.

Thank you for your comments. This study focused on extracting English keywords from scientific publications, and how knowledge development differed across different river basins globally. Other languages were not included for keyword processing. We will justify this limitation in the Discussion section: English remains the most used language for knowledge development across different regions, and many academics with other language backgrounds wrote in English for wider dissemination of findings on their river basins (Ramírez-Castañeda, 2020).

Results Interpretation: You note a strong correlation between knowledge structures and societal impacts, but a weaker link with policy. What might account for this discrepancy? Further analysis or discussion on this point would be beneficial.

Thank you for your comments. The weak link between knowledge and policy can be attributed to the challenge of productive knowledge transfer on decision making, which is commonly studied by research at the science-policy interface (Louder et al., 2021; Nguyen et al., 2017). We will provide additional discussions on this challenge in the Discussion section, noting that:

1) Such challenges arise mainly because that policy and practice decisions are informed by diverse values and beliefs, multiple sources of knowledge, and are shaped by cognitive factors and power dynamics beyond the direct influence of research activities (Hakkarainen et al., 2020; Pitt et al., 2018; Posner & Cvitanovic, 2019).

2) A potential solution will be encouraging the development of "boundary spanners" (Edwards & Meagher, 2020) for effective knowledge transfer between science and practice.

You highlight the importance of interdisciplinary research, particularly in Asian, African, and South American river basins. Could you suggest specific strategies to foster such research in these areas?

Thank you for your comments. We will more thoroughly discuss knowledge development in Asian, African and South American river basins in the revised manuscript, specifically:

1) Development towards an integrated knowledge structure should be most desirable, linking with the Knowledge For Environment (KFE) and the Knowledge For Resource Availability (KFA) patterns. About 15% of the river basins studied in America, Europe and Oceania (e.g., the Amazon River, the Colorado River, the Danube River, the Murray-Darling Basin) provide good examples in achieving a holistic integration of science, society and policy.

2) Recognizing that there are inevitable concerns and interests of these river basins with greater development pressures and inequalities. A more balanced and integrated knowledge development approach could be supported by raising awareness of human impacts on river basins and targeted research fundings that facilitate bridging between science and policy (Jabbour, 2022; Matsumoto et al., 2020).

The concept of "boundary spanning organizations" is introduced as a solution. Further details on the form and mechanisms of these organizations would be valuable.

Thank you for your comments. We will provide additional explanation on the "boundary spanning organization" in the Discussion section, specifically they can be creditable academic organizations for the policy community, individual or groups of scientists or professional consultants. They operate across otherwise disconnected communities (e.g., between policy-makers, local stakeholders and technical experts, between natural and social scientists) to facilitate knowledge and information exchange, and they generally process some power to synthesize different values and insights to facilitate collective sense-making (Bodin, 2017; Stovel & Shaw, 2012).

Overall, this manuscript is well-written, but certain sections could benefit from further linguistic refinement to enhance clarity and flow. This research provides valuable insights

into the structure of scientific knowledge within global river basins and offers constructive strategies for sustainable development. I look forward to your feedback on these comments and the revised manuscript.

Thank you for your comments. We will carefully improve the clarity and flow of language in the revised manuscript.

**References:**

Bodin, Ö. (2017). Collaborative environmental governance: Achieving collective action in social-ecological systems. *Science*, *357*(6352), eaan1114. https://doi.org/10.1126/science.aan1114

Cash, D. W., Clark, W. C., Alcock, F., Dickson, N. M., Eckley, N., Guston, D. H., Jäger, J., & Mitchell, R. B. (2003). Knowledge systems for sustainable development. *Proceedings of the National Academy of Sciences*, *100*(14), 8086-8091. https://doi.org/10.1073/pnas.1231332100

Coccia, M. (2020). The evolution of scientific disciplines in applied sciences: dynamics and empirical properties of experimental physics. *Scientometrics*, *124*(1), 451-487. https://doi.org/https://doi.org/10.1007/s11192-020-03464-y

Edwards, D. M., & Meagher, L. R. (2020). A framework to evaluate the impacts of research on policy and practice: A forestry pilot study. *Forest Policy and Economics*, *114*, 101975. https://doi.org/10.1016/j.forpol.2019.101975

Fortunato, S., Bergstrom, C. T., Börner, K., Evans, J. A., Helbing, D., Milojević, S., Petersen, A. M., Radicchi, F., Sinatra, R., Uzzi, B., Vespignani, A., Waltman, L., Wang, D., & Barabási, A.-L. (2018). Science of science. *Science*, *359*(6379), eaao0185. https://doi.org/10.1126/science.aao0185 %J Science

Hakkarainen, V., Daw, T. M., & Tengö, M. (2020). On the other end of research: exploring community-level knowledge exchanges in small-scale fisheries in Zanzibar. *Sustainability Science*, *15*(1), 281-295. https://doi.org/10.1007/s11625-019-00750-4

Huttenhower, C., Gevers, D., Knight, R., Abubucker, S., Badger, J. H., Chinwalla, A. T., Creasy, H. H., Earl, A. M., FitzGerald, M. G., Fulton, R. S., Giglio, M. G., Hallsworth-Pepin, K., Lobos, E. A., Madupu, R., Magrini, V., Martin, J. C., Mitreva, M., Muzny, D. M., Sodergren, E. J., . . . The Human Microbiome Project, C. (2012). Structure, function and diversity of the healthy human microbiome. *Nature*, *486*(7402), 207-214. https://doi.org/https://doi.org/10.1038/nature11234

Jabbour, J. (2022). *Global sustainability governance: Integrated scientific assessment at a critical inflection point* TU München].

Kim, R. E. (2019). Is Global Governance Fragmented, Polycentric, or Complex? The State of the Art of the Network Approach. *International Studies Review*, *22*(4), 903-931. https://doi.org/10.1093/isr/viz052

Louder, E., Wyborn, C., Cvitanovic, C., & Bednarek, A. T. (2021). A synthesis of the frameworks available to guide evaluations of research impact at the interface of environmental science, policy and practice. *Environmental Science & Policy*, *116*, 258-265. https://doi.org/https://doi.org/10.1016/j.envsci.2020.12.006

Matsumoto, I., Takahashi, Y., Mader, A., Johnson, B., Lopez-Casero, F., Kawai, M., Matsushita, K., & Okayasu, S. (2020). Mapping the Current Understanding of Biodiversity Science–Policy Interfaces. In O. Saito, S. M. Subramanian, S. Hashimoto, & K. Takeuchi (Eds.), *Managing Socio-ecological Production Landscapes and Seascapes for Sustainable Communities in Asia* (pp. 147-170). Springer Singapore. http://link.springer.com/10.1007/978-981-15-1133-2_8

Nguyen, V. M., Young, N., & Cooke, S. J. (2017). A roadmap for knowledge exchange and mobilization research in conservation and natural resource management. *Conservation Biology*, *31*(4), 789-798. https://doi.org/https://doi.org/10.1111/cobi.12857

Pitt, R., Wyborn, C., Page, G., Hutton, J., Sawmy, M. V., Ryan, M., & Gallagher, L. (2018). Wrestling with the complexity of evaluation for organizations at the boundary of science, policy, and practice. *Conservation Biology*, *32*(5), 998-1006. https://doi.org/https://doi.org/10.1111/cobi.13118

Posner, S. M., & Cvitanovic, C. (2019). Evaluating the impacts of boundary-spanning activities at the interface of environmental science and policy: A review of progress and future research needs. *Environmental Science & Policy*, *92*, 141-151. https://doi.org/10.1016/j.envsci.2018.11.006

Ramírez-Castañeda, V. (2020). Disadvantages in preparing and publishing scientific papers caused by the dominance of the English language in science: The case of Colombian researchers in biological sciences. *PLoS ONE*, *15*(9), e0238372. https://doi.org/10.1371/journal.pone.0238372

Sayles, J. S., & Baggio, J. A. (2017). Social–ecological network analysis of scale mismatches in estuary watershed restoration. *Proceedings of the National Academy of Sciences*, *114*(10), E1776-E1785. https://doi.org/https://doi.org/10.1073/pnas.1604405114

Stovel, K., & Shaw, L. (2012). Brokerage. *Annual Review of Sociology*, *38*(Volume 38, 2012), 139-158. https://doi.org/https://doi.org/10.1146/annurev-soc-081309-150054

Von Bertalanffy, L. (1968). General system theory. *New York Magazine*, *41973*(1968), 40.

Wei, Y., & Wu, S. (2022). The gulf of cross-disciplinary research collaborations on global river basins is not narrowed. *Ambio*. https://doi.org/10.1007/s13280-022-01716-0

Wu, S., Wei, Y., & Wang, X. (2021). Structural gaps of water resources knowledge in global river basins. *Hydrol. Earth Syst. Sci.*, *2021*, 1-16. https://doi.org/10.5194/hess-2021-137

Zeng, A., Shen, Z., Zhou, J., Wu, J., Fan, Y., Wang, Y., & Stanley, H. E. (2017). The science of science: from the perspective of complex systems. *Physics Reports*, *714-715*, 1-73. https://doi.org/https://doi.org/10.1016/j.physrep.2017.10.001

---

## Author Response (AR1)

**Responses to reviewers**

Dear Editors and Reviewers,

Thank you very much for your time and effort for reviewing the article initially titled "Impacts of science on society and policy in global river basins" (hess-2024-72). We really appreciate all of your insightful comments.

We have revised the article title to "Impacts of science on society and policy in major river basins globally" to better reflect the scope of the study, and provided a revised version of the manuscript (changes marked in yellow highlights). Hatch patterns have been added to figures with multiple colors for readers with color vision deficiencies.

We have also provided point-by-point responses to each of your comments in black font along with corresponding line numbers (in **bold**) in the revised manuscript below:

**RC1: Comment on hess-2024-72**

The manuscript systematically elaborates on the knowledge system of key river. Among them, the key processes and algorithms has its further selection principles or unique applicability, and there is a detailed introduction. This is very important in the evaluation of knowledge system services and is more conducive to comparison between different studies. However, after overall review, there are still many doubts regarding the following:

Overall Comment:

(1) Formula Standardization

There are descriptive words in formulas 1, 2, and 3 in the manuscript. Please explain the variables and their meanings in a more explicit way (e.g. "where, x is … " in Eq.3), and use a more explicit variable calculation form to present the "median" (Eq.3).

Thank you for your comments. We have refined and added further explanations for Eq.1-3 as follows **(Line 55-65, Line 160-165):**

For any discipline-issue network i:

$$DM_i = \frac{2C_d}{n(n-1)} \qquad \text{(Eq.1)}$$

where $DM_i$ is the Degree of Multidisciplinarity value of a discipline-issue network i, $C_d$ is the total number of existing connections between any issue and discipline d in the network, and n is the total number of d in the network.

For any issue network i:

$$DI_i = \frac{\sum_n C_m}{n} \qquad \text{(Eq.2)}$$

where $DI_i$ is the Degree of Issue-connectivity of an issue network i, $C_m$ is the number of adjacent connections to any specific issue m, and n is the total number of m in the network.

The Sen's slopes (Sen, 1968) were then used to measure the magnitudes of the trends as Eq.3:

$$d_{Sen} = \tilde{d}\left(\frac{x_j - x_i}{j - i}\right) \text{ for } 1 \leq i < j \leq n \qquad\qquad (Eq.3)$$

where $\tilde{d}$ is the median value separating the higher 50% from the lower 50% of the indicator value x in the time series, $i$ and $j$ are adjacent time points, and n is the total number of time points.

(2) Structural Issues

Can the representation of the framework in Section 2 be considered as a preface to the Methods section? Among them, the density of the discipline-issue network and the calculation method for the degree centrality of the issue network are all in the Section 3. It becomes clearer whether they can be merged.

Thank you for your comments. We have integrated Section 2 into the Method section as a new Section 2.1, and consolidated the descriptions of the framework and relevant calculations about the discipline-issue network and the issue network in the new Section 2.1 (**Line 40-90**).

(3) Comprehensive Knowledge Structure

The selection of 72 river basins is mostly typical of river systems in various continents, and is also significantly influenced by human activities. And the information must also be relatively detailed, which is a necessary foundation for this research method. However, can the representativeness of social and policy analysis be highlighted based on existing analysis results? ("Abstract: …Evaluating these structural characteristics against 6 impact indicators on society and policy, over 90% of rivers were found to had knowledge structures that strongly linked to societal impacts whereas only 57% were to the policy…") After all, the title mentions global river basins, but currently the intuitive feeling is to search for conclusions in large rivers influenced by humans, which always feels somewhat inappropriate. Please take above concern into consideration.

Additionally, why are there missing rivers in the North Asian region, such as the Ob River and Yenisei River basins? Will the North Asian rivers, which are relatively low in human activity, affect the relevant conclusions on policy and social impact in the abstract?

Thank you for your comments. The 72 river basins were selected based on those receiving the highest numbers of publications in the WoS database. We chose peer-reviewed publications in the WoS as our data source as it provides consistent, systematic documentation of scientific knowledge development across a broad range of disciplines for a long timeframe. At least one river basin in each of the continents was included for the spatial representativeness of this study. This is clarified in the method section (**Line 115-120**).

However, we agree that there is a potential bias towards large river basins with societal and natural significance to be studied, and some rivers may not be included due to comparatively fewer publications in the WoS. For example, the Lake Baikal catchment was studied, which was a major part of the Yenisei River. We have recognized it as a limitation in the discussion section **(Line 400-405)**.

In addition, we will change our title as "Impacts of science on society and policy in major river basins globally".

(4) Support for key conclusions in the manuscript

The following sentence is an explanation of the key conclusions in the abstract ("Abstract: …over 90% of rivers were found to had knowledge structures that strongly linked to societal impacts whereas only 57% were to the policy"). However, is the R2 the smallest among the 41 basins greater than 0.3, or is the mean of the 41 basins greater than 0.3? The R2 value is indeed a bit low, and the correlation explanation is weak; But it is possible that in such studies, more than 0.3 has already met the interpretive requirements. The manuscript can supplement the general situation of R2 in similar studies and compare the level of 0.3. To enhance the reliability of the conclusions of this article.

" The structural characteristics of the knowledge systems had been strongly linked to the society indicators with over 90% river basins had acceptable regression model fits, but much weaker with the policy indicators as only 41 river basins had two or more linear models that validated the relationships between their knowledge systems and the policy (adjusted R2 > 0.3, statistical significance p < 0.05). "

Thank you for your comments. The $R^2$ values in this study were estimated in each regression model for each river basin, and any models with $R^2$ values smaller than 0.3 were grouped into the 'unclear knowledge-society' or 'unclear knowledge-policy' pattern.

The threshold of 0.3 was selected based on studies in conventional statistical regressions (Ratner, 2009; Royston, 2007), which identified 0.3 to have "weak" explanation power between the knowledge indicators and society/policy indicators. Similar thresholds between 0.2 and 0.3 have also been found by correlations between knowledge, attitudes, and practices regarding environmental problems (Afroz & Ilham, 2020; Alias, 2019). In general, a recent meta-analysis (Hernanda et al., 2023) indicated an acceptable range for correlation levels to be 0.26 to 0.48 across 23 studies published from 1999 to 2022. We have provided this additional justification in the method section (**Line 180-185**).

(5) Section of "Data and code availability"

(Only representing personal opinions) Compared to conclusive summaries, collecting and organizing information and making accurate judgments in the process will be more important. Can the manuscript be supplemented with information about the data or list of statistically

analyzed in the article, in order to facilitate further research development or review during the evaluation process.

Thank you for your comments. We have provided an Appendix with additional explanations for each section detailing the data information and statistical analysis conducted to support the results in the manuscript (**Line 420-490**).

R codes used to generate the results were also commented and deposited in the public repository Github (https://github.com/SLWU423/Code-for-global-river-basin-science-policy-society-impact) for reproduction of the results and further research development (**Line 490-495**).

General Comment:

(1) Image clarity

The text resolution in Figure 2-c is not sufficient to see clearly, and there is overlap with the 0-axis. Is the threshold for "low DM" or "high DI" in the manuscript Line 230~235) divided by the 25th and 75th percentiles in box boundaries?

The resolution in the all figures is not clear, especially in the form of coordinate axis subfigures.

Thank you for your comments. We refrained from introducing additional subjective bias to define a specific threshold value for DM and DI, and considered the comparative values of DM and DI among the 72 river basins. Therefore, the low and high DM and DI were determined by their z-scores:

For any river basin k, and any knowledge, societal, and policy indicator x:

$$x'_k = \frac{x_k - \overline{x_k}}{\sigma_k} \qquad \qquad \text{(Eq.4)}$$

where $x'_k$ is the z-score of any knowledge, societal and policy indicator of $x_k$ , $\overline{x_k}$ is the mean value, and $\sigma_k$ is the standard deviation (**Line 170-175**).

Therefore, we determined the division between 'low' and 'high' scores by the zero value of z-score. A z-score above zero means that the DM or DI value is above the average value for all rivers, and therefore having a 'high' DM or DI. Similarly, a z-score value below zero will be considered having a 'low' DM or DI. This has been clarified in **Line 65-85**.

We have also increased the resolutions and fonts for all figures for improved clarity in the revised manuscript.

(2) Optimization processing of Appendix

The table in the Appendix only requires quantity, and the proportion of 0.00% is the result of omitted accuracy. The number of columns can be changed to reduce pages (Table A1, Table A2).

Thank you for your comments. We have removed the proportion values and reformatted all tables in Appendix A (**Line 420-425**).

**RC2: Comment on hess-2024-72**

This study on the impacts of scientific knowledge development on society and policy within global river basins is both timely and insightful. The framework for measuring knowledge systems through network dimensions of multidisciplinarity and issue-connectivity is commendable. Here are some review comments:

Framework and Methodology: Elaborate on the theoretical underpinnings of your proposed framework and discuss its potential for long-term applicability.

Thank you for your comments. We have consolidated the theoretical underpinning that supports the knowledge network construction in a new Section 2.1 (**Line 40-50**). Specifically: "Built on the Science of Science (SoS) theory (Zeng et al., 2017), a knowledge system is understood as a dynamic system, consisting of knowledge from different disciplines and issues studied, with complex and co-evolving relationships between them, as Latour (1987) described "*knitting, weaving and knotting together into an overarching scientific fabric*" (Latour, 1987; Shi et al., 2015). We adopt a network-based framework to evaluate such interactions (Coccia, 2020; Sayles & Baggio, 2017; Wei et al., 2022; Wu et al., 2021).We characterize the knowledge system as a discipline-issue network, where connections are established between issues and the disciplines used to address the issues (Callon et al., 1983; Noyons, 2001)."

We have also added an additional section in the Discussion section elaborating the implications of the framework (**Line 385-395**). Specifically: "Our network-based framework contributes to advancing the Science of Science (Zeng et al., 2017) and transforming knowledge for more sustainable river basin development. It provides a method to explicitly measure the structure of knowledge as a discipline-issue network system, which guides future knowledge development by identifying explicitly where and what to change or connect between disciplinary knowledge and issues at hand, therefore assisting in more suitable, more precise, and more predictable knowledge development. Moreover, our framework links the structural configurations of knowledge systems with developments in society and policy, thus contribute to better evaluation of research outcomes and action-oriented research for specifying "credible, legitimate, and relevant" in good governance (Cash et al., 2003; Kim, 2019). Finally, this framework will contribute to river basin management by enabling comparisons of knowledge development for river basins with varying management issues of focuses and contexts, thus enables the design of tailored management strategies and co-learning according to different patterns of connections among river basin knowledge, society, and policy development."

Data and Analysis: Consider the inclusion of additional data sources beyond Web of Science, such as conference papers or government reports, to enhance the study's comprehensiveness.

Thank you for your comments. This study focused on the science-driven knowledge development, by using peer-reviewed articles indexed in the Web of Science (WoS) database. The WoS database was chosen because it provides consistent, systematic documentation of knowledge development across a broad range of disciplines for a long timeframe. However, we do acknowledge that use of additional data including conference paper, and government

reports also contributes to the river basin knowledge development, which tends to focus on the practice-driven knowledge. This has been recognized as a limitation in the discussion section (**Line 395-405**).

Address the potential variability in keyword extraction and clustering across different languages and regions.

Thank you for your comments. This study focused on extracting English keywords from scientific publications, and how knowledge development differed across different river basins globally. Other languages were not included for keyword processing.

We have justified this limitation in the Discussion section (**Line 395-405**): English remains the most used language for knowledge development across different regions, and many academics with other language backgrounds wrote in English in studying for wider dissemination of findings on their river basins (Ramírez-Castañeda, 2020).

Results Interpretation: You note a strong correlation between knowledge structures and societal impacts, but a weaker link with policy. What might account for this discrepancy? Further analysis or discussion on this point would be beneficial.

Thank you for your comments. The weak link between knowledge and policy can be attributed to the challenge of productive knowledge transfer on decision making, which is commonly studied by research at the science-policy interface (Louder et al., 2021; Nguyen et al., 2017). We have provided additional discussions under the subtitle of "Challenges at the knowledge-policy interface" in the Discussion section (**Line 355-370**), specifically:

"Over 90% of the river basins had knowledge structures that strongly linked to the society indicators but only 57% of rivers had statistically significant relationships with the policy indicators (Fig. 4). This is closely related to the challenge of knowledge transfer on decision making at the science-policy interface (Louder et al., 2021; Nguyen et al., 2017). Such challenge has been widely recognised as policy and practice decisions are informed by diverse values and beliefs, multiple sources of knowledge, and are shaped by cognitive factors and power dynamics beyond the direct influence of research activities (Hakkarainen et al., 2020; Pitt et al., 2018; Posner & Cvitanovic, 2019). We propose to develop "boundary spanners" as a potential solution (Edwards & Meagher, 2020). These spanners could be creditable academic organizations for the policy community, individual or groups of scientists or professional consultants who facilitate knowledge and information across otherwise disconnected communities and synthesize different values and insights to facilitate collective sense-making (Bodin, 2017; Stovel & Shaw, 2012). They not only can bridge disciplinary silos for natural and social scientists, but more importantly able to coordinate scientists with local stakeholders and policy-makers with different levels of management powers and contexts. Additionally, although beyond the scope of this study, we recognize the interactions between society and policy. In particular, the SO in society indicators and the RU in policy indicators were most

strongly positively correlated ($r = 0.81$, $p < 0.05$) (Fig. B5), which indicates a need to recognise the connections between policy and society development and their spill-over effects on knowledge in future study. ”

You highlight the importance of interdisciplinary research, particularly in Asian, African, and South American river basins. Could you suggest specific strategies to foster such research in these areas?

Thank you for your comments. We have more thoroughly discussed knowledge development in Asian, African and South American river basins in the revised manuscript under the subtitle "Tailored knowledge strategies based on knowledge-society-policy patterns" in the Discussion section (**Line 370-385**). Specifically:

"The integrated knowledge structure was identified to be most desirable, which links with the Knowledge For Environment (KFE) and the Knowledge For Resource Availability (KFA) patterns. Issue-driven knowledge structures were identified to have similar optimized society and policy impacts to the integrated knowledge structure, whereas discipline-driven knowledge structure was not effective in optimizing multiple society and policy indicators at the same time (Fig. 5 and Fig. C1). About 15% of the river basins studied in America, Europe and Oceania (e.g., the Amazon River, the Colorado River, the Danube River, and the Murray-Darling Basin) with integrated knowledge structures demonstrated more balanced impacts on society and policy (Fig. 5). They provide good examples for other river basins in achieving a holistic integration of science, society and policy. On the other hand, river basins with the Knowledge Against Environment (KAE) and the Knowledge Against Resource Availability (KAR) patterns are considered less desirable, as optimizing the current knowledge structure to reduce the negative environmental impacts or improving resource availabilities would be traded off with socio-economic development and governance capacities (Fig. 5). Rivers with fragmented knowledge structures comprising 35% of the river basins studied, mostly in Asia, Africa, and South America were most prone to these impact patterns (Fig. 3). It reflects the inevitable concerns and interests of these river basins with greater development pressures and inequalities. A more balanced and integrated knowledge development approach could be supported by raising awareness of human impacts on river basins, and targeted research fundings that facilitate bridging between science and policy (Jabbour, 2022; Matsumoto et al., 2020)."

The concept of "boundary spanning organizations" is introduced as a solution. Further details on the form and mechanisms of these organizations would be valuable.

Thank you for your comments. We have provided additional explanation on the "boundary spanning organization" in the Discussion section (**Line 360-365**). Specifically:

"We propose to develop "boundary spanners" as a potential solution (Edwards & Meagher, 2020). These spanners could be creditable academic organizations for the policy community,

individual or groups of scientists or professional consultants who facilitate knowledge and information across otherwise disconnected communities and synthesize different values and insights to facilitate collective sense-making (Bodin, 2017; Stovel & Shaw, 2012). They not only can bridge disciplinary silos for natural and social scientists, but more importantly able to coordinate scientists with local stakeholders and policy-makers with different levels of management power and contexts."

Overall, this manuscript is well-written, but certain sections could benefit from further linguistic refinement to enhance clarity and flow. This research provides valuable insights into the structure of scientific knowledge within global river basins and offers constructive strategies for sustainable development. I look forward to your feedback on these comments and the revised manuscript.

Thank you for your comments. We will carefully improve the clarity and flow of language in the revised manuscript.

**References:**

Afroz, N., & Ilham, Z. (2020). Assessment of Knowledge, Attitude and Practice of University Students towards Sustainable Development Goals (SDGs). *The Journal of Indonesia Sustainable Development Planning*, *1*(1), 31-44. https://doi.org/10.46456/jisdep.v1i1.12

Alias, N. A. (2019). Correlation between knowledge, attitude, and behavior towards river pollution. *International Journal of Modern Trends in Social Sciences*, *2*(9), 31-38.

Bodin, Ö. (2017). Collaborative environmental governance: Achieving collective action in social-ecological systems. *Science*, *357*(6352), eaan1114. https://doi.org/10.1126/science.aan1114

Callon, M., Courtial, J.-P., Turner, W. A., & Bauin, S. (1983). From translations to problematic networks: An introduction to co-word analysis. *Information (International Social Science Council)*, *22*(2), 191-235. https://doi.org/https://doi.org/10.1177/053901883022002003

Cash, D. W., Clark, W. C., Alcock, F., Dickson, N. M., Eckley, N., Guston, D. H., Jäger, J., & Mitchell, R. B. (2003). Knowledge systems for sustainable development. *Proceedings of the National Academy of Sciences*, *100*(14), 8086-8091. https://doi.org/10.1073/pnas.1231332100

Coccia, M. (2020). The evolution of scientific disciplines in applied sciences: dynamics and empirical properties of experimental physics. *Scientometrics*, *124*(1), 451-487. https://doi.org/https://doi.org/10.1007/s11192-020-03464-y

Edwards, D. M., & Meagher, L. R. (2020). A framework to evaluate the impacts of research on policy and practice: A forestry pilot study. *Forest Policy and Economics*, *114*, 101975. https://doi.org/10.1016/j.forpol.2019.101975

Hakkarainen, V., Daw, T. M., & Tengö, M. (2020). On the other end of research: exploring community-level knowledge exchanges in small-scale fisheries in Zanzibar. *Sustainability Science*, *15*(1), 281-295. https://doi.org/10.1007/s11625-019-00750-4

Hernanda, T., Absori, Azhari, A. F., Wardiono, K., & Arlinwibowo, J. (2023). Relationship Between Knowledge and Affection for the Environment: A Meta-Analysis. *European*

*Journal of Educational Research*, *12*(2), 1071-1084. https://doi.org/10.12973/eu-jer.12.2.1069

Jabbour, J. (2022). *Global sustainability governance: Integrated scientific assessment at a critical inflection point* TU München].

Kim, R. E. (2019). Is Global Governance Fragmented, Polycentric, or Complex? The State of the Art of the Network Approach. *International Studies Review*, *22*(4), 903-931. https://doi.org/10.1093/isr/viz052

Latour, B. (1987). *Science in action: How to follow scientists and engineers through society*. Harvard university press.

Louder, E., Wyborn, C., Cvitanovic, C., & Bednarek, A. T. (2021). A synthesis of the frameworks available to guide evaluations of research impact at the interface of environmental science, policy and practice. *Environmental Science & Policy*, *116*, 258-265. https://doi.org/https://doi.org/10.1016/j.envsci.2020.12.006

Matsumoto, I., Takahashi, Y., Mader, A., Johnson, B., Lopez-Casero, F., Kawai, M., Matsushita, K., & Okayasu, S. (2020). Mapping the Current Understanding of Biodiversity Science–Policy Interfaces. In O. Saito, S. M. Subramanian, S. Hashimoto, & K. Takeuchi (Eds.), *Managing Socio-ecological Production Landscapes and Seascapes for Sustainable Communities in Asia* (pp. 147-170). Springer Singapore. http://link.springer.com/10.1007/978-981-15-1133-2_8

Nguyen, V. M., Young, N., & Cooke, S. J. (2017). A roadmap for knowledge exchange and mobilization research in conservation and natural resource management. *Conservation Biology*, *31*(4), 789-798. https://doi.org/https://doi.org/10.1111/cobi.12857

Noyons, E. (2001). Bibliometric mapping of science in a science policy context.

Pitt, R., Wyborn, C., Page, G., Hutton, J., Sawmy, M. V., Ryan, M., & Gallagher, L. (2018). Wrestling with the complexity of evaluation for organizations at the boundary of science, policy, and practice. *Conservation Biology*, *32*(5), 998-1006. https://doi.org/https://doi.org/10.1111/cobi.13118

Posner, S. M., & Cvitanovic, C. (2019). Evaluating the impacts of boundary-spanning activities at the interface of environmental science and policy: A review of progress and future research needs. *Environmental Science & Policy*, *92*, 141-151. https://doi.org/10.1016/j.envsci.2018.11.006

Ramírez-Castañeda, V. (2020). Disadvantages in preparing and publishing scientific papers caused by the dominance of the English language in science: The case of Colombian researchers in biological sciences. *PLoS ONE*, *15*(9), e0238372. https://doi.org/10.1371/journal.pone.0238372

Ratner, B. (2009). The correlation coefficient: Its values range between +1/−1, or do they? *Journal of Targeting, Measurement and Analysis for Marketing*, *17*(2), 139-142. https://doi.org/10.1057/jt.2009.5

Royston, P. (2007). Profile Likelihood for Estimation and Confidence Intervals. *The Stata Journal*, *7*(3), 376-387. https://doi.org/10.1177/1536867x0700700305

Sayles, J. S., & Baggio, J. A. (2017). Social–ecological network analysis of scale mismatches in estuary watershed restoration. *Proceedings of the National Academy of Sciences*, *114*(10), E1776-E1785. https://doi.org/https://doi.org/10.1073/pnas.1604405114

Sen, P. K. (1968). Estimates of the Regression Coefficient Based on Kendall's Tau. *Journal of the American Statistical Association*, *63*(324), 1379-1389. https://doi.org/10.1080/01621459.1968.10480934

Shi, F., Foster, J. G., & Evans, J. A. (2015). Weaving the fabric of science: Dynamic network models of science's unfolding structure. *Social Networks*, *43*, 73-85. https://doi.org/https://doi.org/10.1016/j.socnet.2015.02.006

Stovel, K., & Shaw, L. (2012). Brokerage. *Annual Review of Sociology, 38*(Volume 38, 2012), 139-158. https://doi.org/https://doi.org/10.1146/annurev-soc-081309-150054

Wei, Y., Wu, S., Lu, Z., Wang, X., Wu, X., Xu, L., & Sivapalan, M. (2022). Ageing Knowledge Structure in Global River Basins [Brief Research Report]. *Frontiers in Environmental Science, 10*. https://doi.org/10.3389/fenvs.2022.821342

Wu, S., Wei, Y., & Wang, X. (2021). Structural gaps of water resources knowledge in global river basins. *Hydrol. Earth Syst. Sci., 2021*, 1-16. https://doi.org/10.5194/hess-2021-137

Zeng, A., Shen, Z., Zhou, J., Wu, J., Fan, Y., Wang, Y., & Stanley, H. E. (2017). The science of science: from the perspective of complex systems. *Physics Reports, 714-715*, 1-73. https://doi.org/https://doi.org/10.1016/j.physrep.2017.10.001